# Varying strength of selection contributes to the intragenomic diversity of rRNA genes

Daniel Sultanov [1] & Andreas Hochwagen [1] ✉

Ribosome biogenesis in eukaryotes is supported by hundreds of ribosomal RNA (rRNA) gene copies that are encoded in the ribosomal DNA (rDNA). The multiple copies of rRNA genes are thought to have low sequence diversity within one species. Here, we present species-wide rDNA sequence analysis in *Saccharomyces cerevisiae* that challenges this view. We show that rDNA copies in this yeast are heterogeneous, both among and within isolates, and that many variants avoided fixation or elimination over evolutionary time. The sequence diversity landscape across the rDNA shows clear functional stratification, suggesting different copy-number thresholds for selection that contribute to rDNA diversity. Notably, nucleotide variants in the most conserved rDNA regions are sufficiently deleterious to exhibit signatures of purifying selection even when present in only a small fraction of rRNA gene copies. Our results portray a complex evolutionary landscape that shapes rDNA sequence diversity within a single species and reveal unexpectedly strong purifying selection of multi-copy genes.

The ribosomal DNA (rDNA) codes for the ribosomal RNAs (rRNAs), the universal RNA components of the ribosome, and therefore is essential for protein translation and viability in every species. Each rDNA copy of *S. cerevisiae* consists of several structural elements, including the coding sequences of the four rRNAs (*5S*, *5.8S*, *18S*, *25S*), *ETS* and *ITS* elements, which are essential for rRNA maturation by interacting with small nucleolar RNAs and nucleases[1], and non-transcribed sequences (*NTS1* and *NTS2*), which contain promoters and the origin of replication[2–4]. Most eukaryotic genomes harbor dozens or hundreds of rDNA copies arranged in one or multiple loci[5]. It is well established that rDNA copy numbers vary widely even among individuals of a single species, contributing to phenotypic diversity[6–9]. By contrast, how much rDNA sequences vary within a genome remains poorly understood.

rDNA sequence variation has been mostly studied in light of the concerted evolution model, which proposes that rDNA copies share derived changes, both on the level of genomes and species, as a consequence of mechanisms such as gene conversion and repeated unequal crossing over[10–13]. High rates of these mechanisms are expected to result in (1) high homogeneity of rDNA copies within a genome with only a few sequence variants, (2) little difference in the distribution of variation among *rRNA*, *ETS*, *ITS*, and *NTS* elements, and

(3) the rapid elimination or fixation of new variants across all rDNA copies[11,12]. However, the assumption of high rates of such mechanisms, and hence these predictions, were based on very limited sequencing information with low sequencing depth from only a few individuals. Indeed, recent studies have detected a substantial number of within-genome rDNA variants across diverse phyla, including fungi[14–16], invertebrates[17–20], plants[21], and mammals, including humans[22–24]. Moreover, several rDNA sequence variants in mammals and insects appear to have persisted over extended evolutionary times[20,24], conflicting with the postulated rapid fixation of variants.

On the other hand, if different modes of selection contribute to rDNA sequence evolution (with low or moderate rates of concerted evolution), then one expects (1) higher intragenomic rDNA variation, (2) marked differences in variation among different rDNA elements, depending on the phenotypic effect of a given sequence variant, (3) a range of frequencies for different variants proportional to their effects on fitness as a result of directional selection, and (4) randomly fluctuating frequencies of variants that are not under selection. Interestingly, the sequences of several multi-copy gene families, including the major histocompatibility complex genes and the highly conserved histone and ubiquitin genes are individually under strong purifying selection[25–28], demonstrating how selection might act on multi-copy

[1]Department of Biology, New York University, New York, NY 10003, USA. ✉e-mail: andi@nyu.edu

genes alongside concerted events. In addition, diversifying selection has maintained multiple variants over long periods of time in the repeated globin genes of mammals[29–31].

Here, we take advantage of the high sequencing coverage of the 1002 Yeast Genomes Project[32] to quantitatively assess within- and between-genome sequence heterogeneity and determine the evolutionary landscape of the rDNA in *Saccharomyces cerevisiae*. Together with comparative and structural information of the ribosome, these data allow us to identify signatures of stringent purifying selection on a species scale and define regions of constrained and permissive variation of rRNA in different regions of the ribosome.

## Results

### Substantial rDNA sequence diversity

To investigate the level of sequence diversity across rDNA elements, we analyzed rDNA sequences of the 1002 Yeast Genomes Project, which comprises deep-sequencing data (-200-fold mean coverage) of 918 *S. cerevisiae* isolates from 21 ecological niches[32], including 133 haploid and 694 diploid isolates, as well as 91 isolates with higher ploidies. We used LoFreq[33] to identify rDNA sequences that differed from the rDNA prototype of the S288c reference strain and considered the alternative sequences to be "rDNA variants" if more than 0.5% of the rDNA reads contained such sequences (Fig. 1a, Supplementary Fig. 1b). In addition, because of the variable number of rDNA copies in different isolates (Fig. 1b), we used a frequency cutoff corresponding to one rDNA copy for considering variants in each isolate (see "Methods"). We refer to the relative abundance of a variant among all rDNA reads as "intragenomic variant frequency" (iVF) (Fig. 1a, Supplementary Fig. 1).

Benchmarking with a test set showed that our pipeline efficiently recovered low-frequency variants down to 0.5% iVF (>97% recovery; Supplementary Fig. 2 and "Methods"), and thus allows detection of variants that are present in only one or a few rDNA copies within a genome. Additional filtering of long homopolymer stretches and low-quality variants further ensured a low number of false-positive calls (Supplementary Fig. 2).

We observed high rDNA sequence variation, both within individual genomes (intragenomic) and between different isolates (intergenomic; Fig. 1c, Supplementary Fig. 3). Within genomes, a median of 13 variants per isolate co-existed with S288c reference sequences (Fig. 1d). In the highly conserved rRNA genes, the median was two variants per isolate, including one case with 40 low-frequency variants (isolate name from ref. 32; "BAM"; Supplementary Fig. 4).

Intergenomic variation was similarly common (different colors; Fig. 1c), and the number of variants per genome only weakly correlated with the total number of rDNA copies per isolate (Supplementary Fig. 5a). Notably, almost half (44%) of the variants were shared across multiple isolates (Supplementary Fig. 5b) but occurred at various iVFs (i.e., in different proportions of rDNA copies) and thus represented intragenomic variant frequency polymorphisms (iVFPs; Fig. 1a). We note that since clonal populations were sequenced, and every isolate was analyzed separately, every iVF represents the frequency of a variant within a genome of a particular isolate and not the population-wide genomic frequency.

Among all isolates, we detected 2129 variants across 1931 polymorphic sites, indicating that most sites exhibited only one type of sequence change. The majority of variants were single-nucleotide variants (SNVs) and short indels (<3 bp; range: 1–24 bp; Fig. 1e, Supplementary Fig. 6). Across the entire data set, we observed a total of 21,562 iVFPs with frequencies from 0.5 to 100% (Fig. 1e; Supplementary Data 1), with most polymorphic sites occurring in the *NTS* regions (Supplementary Fig. 7, Supplementary Data 2).

Closer inspection of the functional rDNA elements revealed that at least 13% of nucleotide positions in each element were polymorphic (Fig. 1e). The number of variants correlated strongly with the length of

each element (Supplementary Fig. 8), which is consistent with random mutation accumulation, but variants were least likely to be shared among isolates if they occurred in the rRNA genes compared to *ETS* + *ITS* and *NTS* elements (median: 1 (*rRNA*) vs. 2 (*NTS*; *ETS* + *ITS*) isolates per variant, $P < 2 \times 10^{-16}$, pairwise Wilcoxon rank-sum test with Benjamini–Hochberg correction; Supplementary Fig. 9). As rRNAs fold into complex structures via canonical and non-canonical base pairings, deleterious effects of base changes most likely prevented their evolutionary persistence. By contrast, the *NTS1/2* elements contain many spacer sequences and thus are more permissible for variation[25]. The difference in selective pressure was also apparent when iVFPs were empirically separated into three categories based on iVFs–low (<5%), mid (from 5% to 95%), and high (≥95%; Fig. 1e, f). We predominantly observed variants with mid- and high iVFs in regulatory sequences (*NTS1/2*, *ETS1/2*, *ITS1*) and low iVFs in the rRNA genes (Fig. 1e). Nevertheless, 25% (232) of variants in *25S*, *18S*, and *5.8S* had iVFs above 5%, indicating even these conserved elements contain positions with relaxed sequence constraints for variation.

When compared to the S288c reference rDNA, iVFPs followed a U-shape distribution regardless of the isolates' ploidy (Fig. 1f, and Supplementary Fig. 10), with peaks at low (below 5%) or high (above 95%) iVFs. For the *NTS* regions, the highest peak corresponded to very high iVFs (≥99%) and reflects virtually fixed variants among rDNA copies. None of these variants were shared across all 918 isolates (Supplementary Fig. 11), indicating that the peak at iVF ≥ 99% was not inflated by the choice of S288c as the reference. However, we did observe five variants, all in the *NTS* regions, that were shared across over 500 isolates and were responsible for 2989/6507 (46%) iVFPs with iVF ≥ 99% in *NTS*, explaining some of the skews in the distribution. In contrast, the *ETS* and *ITS* regions ("ETS + ITS") exhibited equally distributed peaks at low and high iVFs, pointing to greater selective pressure in these elements due to their importance in rRNA processing and maturation[1]. In the *rRNA*-encoding elements, iVFPs with low iVFs dominated, presumably as a result of stringent selection.

The U-shaped distribution of iVFPs held within each ecological niche, with interesting outliers (Fig. 1g). For instance, dairy isolates exhibited a greater variation of iVFs in the rRNA-encoding sequences, and isolates involved in bioethanol production exhibited a peak with iVFs around 50% in the *NTS* regions, perhaps reflecting recent hybridization. Both groups were large (27 isolates) and consisted primarily of heterozygous diploid isolates (>92%), arguing against outlier effects from unusual ploidies.

Since multiple variants can arise in the same rDNA copy, we asked whether pairs of variants repeatedly co-occurred. Short-read sequencing data only detects co-occurring variants when they are present in the same short read. However, we can infer that two or more variants may occur in the same rDNA copy if they exhibit consistently similar iVFs across multiple isolates (e.g., variants X and Y both had iVF ~ 25% in isolate A and ~8% in isolate B; see "Methods"). We observed 2864 such coupled variant pairs within and between rDNA elements (Supplementary Data 3, 4; Supplementary Figs. 12, 13). The majority (71%) of those variants had iVF ≥ 80%, which unambiguously indicated their co-occurrence in the same set of rDNA copies. Occasionally, we also observed variants within the same sequencing read pair. One notable example included five co-occurring SNVs within a stretch of 38 nucleotides in *25S* of several industrial strains (Supplementary Fig. 14).

Taken together, these data reveal pervasive heterogeneity in rDNA sequences within and between genomes of the same species and show non-uniform functional sequence constraints.

### Stratification by environment

Previous work demonstrated that domestication strongly influences mutation accumulation, with wild *S. cerevisiae* isolates accumulating single-nucleotide polymorphisms and domesticated strains predominantly utilizing changes in gene copy number to increase fitness[32].

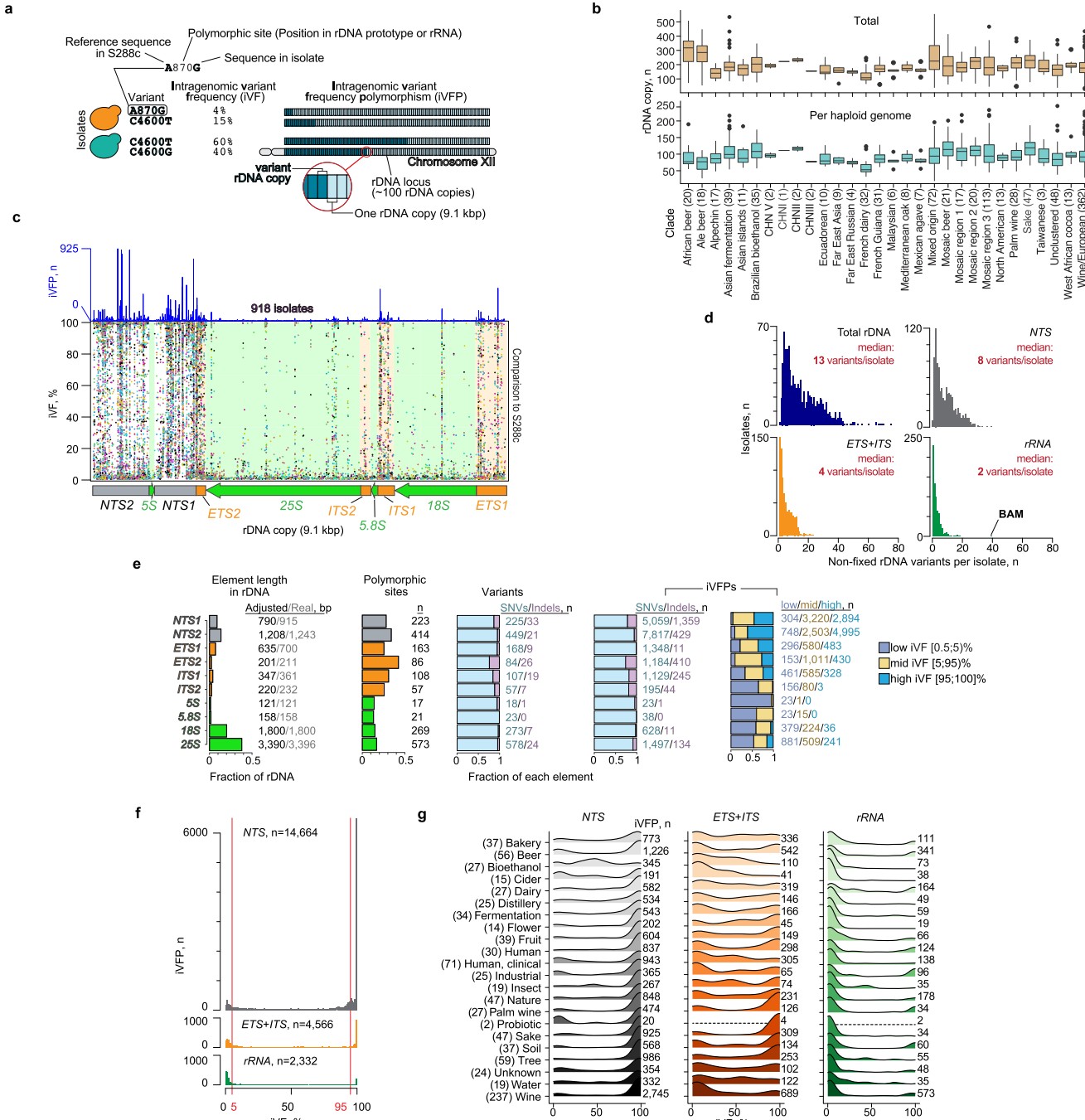

**Fig. 1 | rDNA sequence variants across 918 _S. cerevisiae_ isolates. a** Variant annotation. Variants are defined as nucleotide differences from the S288c prototype (light blue). Intragenomic variant frequency (iVF) is the percentage of rDNA reads with a variant sequence within an isolate's genome and is proportional to variant rDNA copies within a genome. Intragenomic variant frequency polymorphisms (iVFP) are shown in dark blue. One isolate can have multiple sequences at a position and thus the sum of the iVFPs can be more than the total number of isolates. **b** rDNA copy number distribution across isolates grouped by clade. Numbers in brackets indicate the number of isolates in each clade. The center line is the median, box limits are 25th and 75th percentiles, and the whiskers extend to ±1.5xIQR (interquartile range). Dots show isolates with values outside the whiskers. **c** iVFPs (dots) compared to the reference S288c rDNA prototype are plotted against one rDNA copy. Colors reflect different isolates (some isolates have same colors because of palette constraints). An rDNA copy is shown. The histogram above

shows the number of iVFPs per position across all isolates. **d** Distribution of the number of non-fixed rDNA variants (present in a maximum of all but one rDNA copy in a genome; see "Methods"). Isolate "BAM" is indicated. **e** Summary statistics. Element length in rDNA—the relative length of each element in the rDNA; Polymorphic sites—the fraction of all variable sites; Variants—the number of observed variants and fraction of single-nucleotide variants (SNVs) and indels; iVFP—the number of iVFPs across all isolates and the fraction SNVs and indels. The numbers indicate absolute values. The adjusted element lengths are due to filtering (see "Methods"). **f** Distribution of iVFPs by iVF in _NTS_, _ETS+ITS_, and _rRNA_. **g** Same analysis as (**f**) but separated by ecological niche. Brackets—the number of isolates in each niche; the number of iVFPs is shown on the right of each distribution. The missing distributions in the Probiotic niche are due to the low number of iVFPs. Source data are provided as a Source data file.

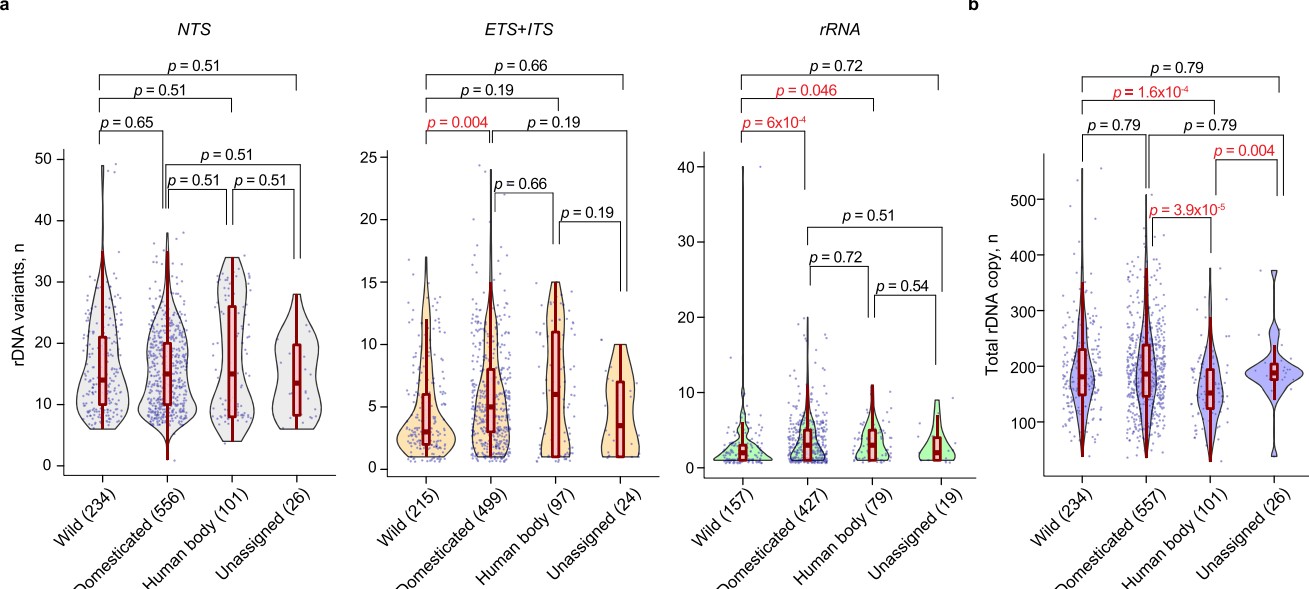

**Fig. 2 | rDNA variation is stratified by habitat. a** Distribution of the number of rDNA variants across different habitats from ref. 34 grouped by rDNA element. Significance test: two-sided Wilcoxon rank-sum test adjusted for multiple hypothesis testing using the Benjamini–Hochberg correction. For *ETS+ITS*, wild vs domesticated: W = 45,070, $p = 0.004$, location shift = −1, 95% confidence interval (CI) [−1,0]. For rRNA, wild vs domesticated: W = 26,653, $p = 6 \times 10^{-4}$, location shift = −1, 95% CI [−1, −6×10⁻⁵]; wild vs human body: W = 5038, $p = 0.046$, location shift = −6.2×10⁻⁶, 95% CI [−1, 0]. **b** Distribution of total rDNA copy numbers across the

habitats. Significance test: two-sided Wilcoxon rank-sum test adjusted for multiple hypothesis testing using the Benjamini–Hochberg correction. For human body vs wild: W = 8534.5, $p = 1.6 \times 10^{-4}$, location shift = −30, 95% CI [−44,−16]; for human body vs domesticated: W = 20,162, $p = 3.9 \times 10^{-5}$, location shift = −32, 95% CI [−46,−18]; for human body vs unknown: W = 800.5, $p = 0.004$, location shift = −34, 95% CI [−54, −13]. For **a** and **b**, each dot is an isolate, the center line in the red box plot is the median, box limits are 25th and 75th percentiles, and the whiskers extend to ±1.5xIQR. Source data are provided as a Source data file.

To investigate whether domestication also affected variant accumulation in the rDNA, we took advantage of the recent classification of the yeast isolates based on their habitat—"domesticated", "wild", "human body", or unassigned[34] (Fig. 2). iVFP distribution was unaffected by habitat (Supplementary Fig. 15), but we observed a significant enrichment of variants in the *ETS + ITS* elements and *rRNA* of domesticated isolates compared to the wild group (median: 5 vs. 3 variants in *ETS + ITS*, $p = 0.004$; median: 3 vs. 2 variants in *rRNA*, $p = 6 \times 10^{-4}$; Wilcoxon rank-sum test with Benjamini–Hochberg correction; Fig. 2a). Copy number estimation further revealed that human body-associated isolates had consistently lower number of total rDNA copies (median: 152 copies) than wild (median: 181 copies, $p = 1.6 \times 10^{-4}$), domesticated (median: 186 copies, $p = 3.9 \times 10^{-5}$), and unassigned (median: 188 copies, $p = 0.004$) isolates (Wilcoxon rank-sum test with Benjamini–Hochberg correction; Fig. 2b). Therefore, the accumulation of rRNA gene variants and rDNA copy numbers in yeast are stratified by growth environment.

## Persistence of variants

Genome heterozygosity increases over time by random mutation accumulation and outcrossing, whereas mitotic recombination and inbreeding can lead to partial or complete loss of heterozygosity. To further investigate the characteristics of mutation accumulation in rDNA, we compared rDNA sequence diversity with the level of heterozygosity of non-rDNA sequences for 781 non-haploid isolates. Since most of the natural isolates were diploid while polyploid isolates were enriched in human-related environments[32], we analyzed these two groups separately.

We found that the number of rDNA variants in each isolate correlated well with the number of heterozygous single-nucleotide polymorphisms in the rest of the genome, regardless of ploidy (Fig. 3a, Supplementary Fig. 16). Conversely, the number of rDNA variants in diploid isolates negatively correlated with the extent of loss of heterozygosity (Fig. 3b). In line with this observation, fully homozygous diploid isolates exhibited lower variant accumulation in the rDNA

(median: 6 variants; Fig. 3c) compared to heterozygous isolates (median: 18 variants; $p = 1.3 \times 10^{-46}$, Wilcoxon rank-sum test), even though both groups had virtually the same median number of rDNA copies (180 vs 184 copies, Wilcoxon rank-sum test; Supplementary Fig. 17). Therefore, rDNA variant accumulation correlates with the rest of the genome.

To investigate how long iVFs persist across the phylogenetic tree, we determined the distribution of the topmost common variants in each element across the clades defined by the 1002 Yeast Genome Project (Fig. 4). We repeatedly found that the same variants presented within clades and across related clades with different iVFPs. The most parsimonious explanation for these widespread variants is that they represent single-origin mutation events that emerged early in the isolates' evolution and were retained after the groups diverged. For example, the estimate for the domestication time of wine strains is about 1.5 thousand years ago[32] and the wine isolates share the same variants at different frequencies both within and between clades. Similarly, some variants in the Wine/European clade were stratified by subclades, with a wide distribution of iVFPs (Supplementary Fig. 18). For the variants in *rRNA*, the wide range of intragenomic frequencies both within and between clades implies that these variants are likely neutral or represent standing variation, with modest fitness effects under some stress conditions (Supplementary Figs. 19, 20). These data demonstrate that genetic diversity in the rDNA can be maintained over extended periods of evolutionary time, arguing against rapid sequence homogenization of rDNA copies.

## Selection profile of the yeast ribosome

The large number of variants with varying frequencies provided an opportunity to quantitatively investigate the power of selection on the rDNA. The ribosome is one of the most ancient molecular machines that likely appeared first as a catalytic piece of RNA that subsequently grew in complexity[35]. It is comprised of a large subunit (LSU) with the 25S, 5.8S, and 5S rRNAs, and a small subunit (SSU) with the 18S rRNA.

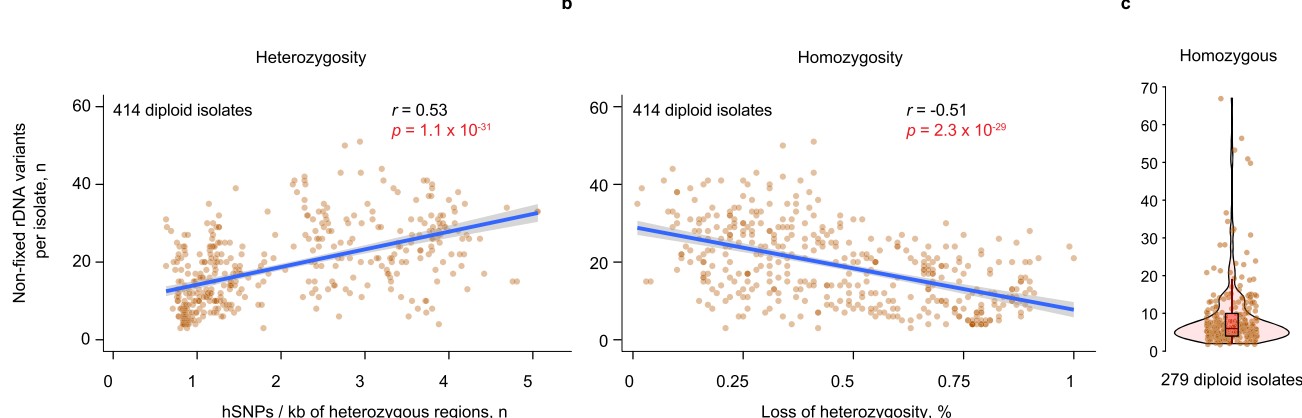

**Fig. 3 | Relationship between sequence diversity of the rDNA and the rest of the genome in diploid isolates.** Correlation between the number of rDNA variants and **a** the number of heterozygous SNPs per heterozygous region of the genome (excluding rDNA)[32] or **b** the loss of heterozygosity per heterozygous isolate. Pearson correlation coefficient R is shown. Each dot represents an isolate. The regression line (blue) is plotted with linear regression model. The gray shading is a 95% CI. Significance test: two-sided Pearson's product-moment correlation. For **a**, t(412) =

12.77, $p = 1.1 \times 10^{-31}$, $r = 0.53$, 95% CI [0.46, 0.6]; for **b**, t(412) = −12.18, $p = 2.3 \times 10^{-29}$, $r$ = −0.51, 95% CI [−0.58, −0.44]. **c** The number of rDNA variants per homozygous isolate (violin plot; red box plot inside shows the median). Each dot is an isolate. The center line in the red box plot is the median, box limits are 25th and 75th percentiles, and the whiskers extend to ±1.5xIQR. Source data are provided as a Source data file.

Comparative structural studies using high-resolution structures of archaeal and bacterial ribosomes showed that the most conserved sequences are localized in the innermost core of the subunits[36,37]. When analyzed in a series of 10 Å-wide concentric shells, the level of rRNA conservation gradually drops from the core outward and plateaus in the outer shells, suggesting that the evolutionarily youngest rRNA elements are located closer to the solvent-exposed surfaces of the ribosome[37]. We adapted this approach and used the rRNA nucleotide distribution data across the shells from ref. 38 to analyze the spatial distribution of the detected variants in the *S. cerevisiae* 80S ribosome (Fig. 5a and Supplementary Data 5).

In line with the evolutionary analyses, iVFPs were significantly skewed toward the outer shells of the ribosome (LSU: low iVF $p = 0.026$, mid iVF $p = 0.013$, high iVF $p = 0.013$; SSU: low iVF $p = 0.015$, mid iVF $p = 0.006$, high iVF $p = 0.006$, Wilcoxon rank-sum exact test with Benjamini–Hochberg correction). As expected, variants with iVFs ≥ 5% were absent in shell 1 of the LSU, which contains the peptidyltransferase center, the highly conserved catalytic core of the ribosome. Variants were similarly depleted in shell 1 of the SSU. This is the region of the highly conserved decoding center, which is essential for the fidelity of codon-anticodon interactions and controlling tRNA selection[39]. This pattern is also consistent with a previously reported conservation pattern in the rRNA of other eukaryotic species[24]. The depletion of higher-frequency variants (no mid- and high iVFPs variants in the LSU; no high iVFPs in the SSU and only one variant with iVF 7.1%) in the core shells likely reflects mutation-selection balance, whereby newly arising variants are generally highly deleterious and countered by stringent purifying selection. Several mid- and high-frequency variants were localized to the second most inner shell 2. In the LSU, six iVFPs had intragenomic frequencies above 5% in the shell, and three of them interact with ribosomal proteins (RPs) (Supplementary Data 5). Notably, we observed no significant differences in iVFs between rRNA variants localized in binding interfaces with RPs and variants outside of those interfaces (Supplementary Fig. 21), arguing that nucleotides involved in RP contacts do not experience significantly different selective forces. In the SSU, seven iVFPs in shell 2 had intragenomic frequencies above 5%, three of which were located in or near functionally important ribosomal regions (see "Variants in functional regions" below), suggesting some of them might be neutral or under positive selection. On the other hand, the outermost regions largely contain rRNA expansion segments (ES; Supplementary Fig. 22),

which vary greatly in their sequence and length between eukaryotes[40–42], and thus are more permissible for mutations (Fig. 5b).

To estimate the strength of purifying selection, we analyzed variants in the conserved nucleotide elements (CNEs), which have virtually the same sequences (>90% identity) across all available eukaryotic 25S/28S rRNA genes[43]. This high evolutionary conservation implies that any variants observed in CNEs can be considered highly deleterious, and thus allowed us to probe for the existence of copy-number thresholds for selection. Mid- and high-frequency variants were almost absent in the CNEs, with the majority of identified CNE variants (135/149; 91%) occurring at iVFs below 6% (Fig. 5c, d; Supplementary Data 5). These low iVFs suggest that mutations in the CNEs are sufficiently deleterious to be selected against even at very low rDNA copy numbers. In fact, even iVFPs with iVFs down to below 1% were significantly underrepresented in CNEs compared to the rest of the 25S sequence ($p = 1.3 \times 10^{-8}$, Fisher's exact test with Benjamini–Hochberg correction; Fig. 5e). Such low iVFPs correspond to only one or a few rDNA copies per cell (Supplementary Fig. 23). In some cases, the depletion of variants with iVF < 1% in CNEs may be due to the additive presence of other deleterious *rRNA* mutations in the same genome that shifts the cumulative iVF to >1% (Supplementary Fig. 24; median: 3%). Nevertheless, these results indicate that purifying selection acts on variants present at a frequency of as low as 1%, and thus near the level of individual rDNA copies.

## Variants in functional regions

A total of 11 CNE variants were observed at higher iVFs (≥6%) (Fig. 5d). Strikingly, eight of these localized along the GTPase-associated center (GAC; Fig. 6a, b), a highly conserved 57 nucleotide-long system of hairpins located in domain II of the 25S rRNA that interacts with translational GTPases during protein synthesis[44]. The eight variants were distributed among nine isolates at various frequencies (Fig. 6c). Closely related isolates ("CFH", "CBR", and "CFG") shared the same variant C1248U (25S rRNA nomenclature; iVFs 20–33%), suggesting a single origin, while the same position also exhibited a different variant sequence in isolate "CFA" (C1248A).

Intriguingly, two variants, C1248U and G1237A, were found in the same isolate ("CFH"), both at 28% frequency. We retrieved sequence reads spanning both positions in the GAC region of the isolate and found that the variants were present in mutually exclusive gene copies, therefore contributing to three distinct forms of 25S in "CFH" (Fig. 6d).

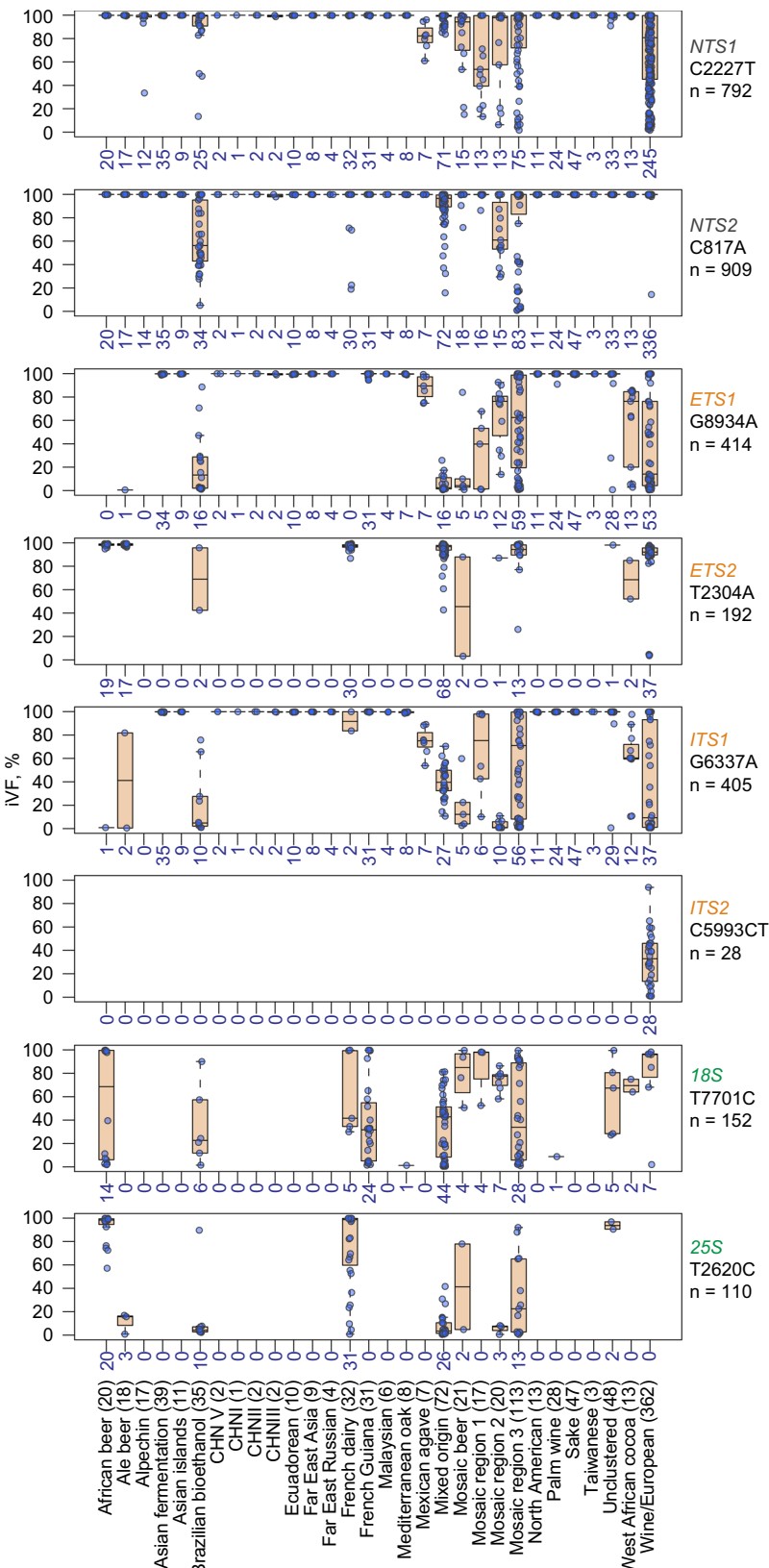

**Fig. 4 | Box plot distribution of the topmost abundant rDNA variants in each rDNA element across clades.** Numbers indicate both the number of isolates with a variant per clade and the number of iVFPs of a specific variant, since one isolate can only have one iVFP of each variant per plot (also shown as dots). The center line in the red box plot is the median, box limits are 25th and 75th percentiles, and the whiskers extend to ±1.5xIQR. Source data are provided as a Source data file.

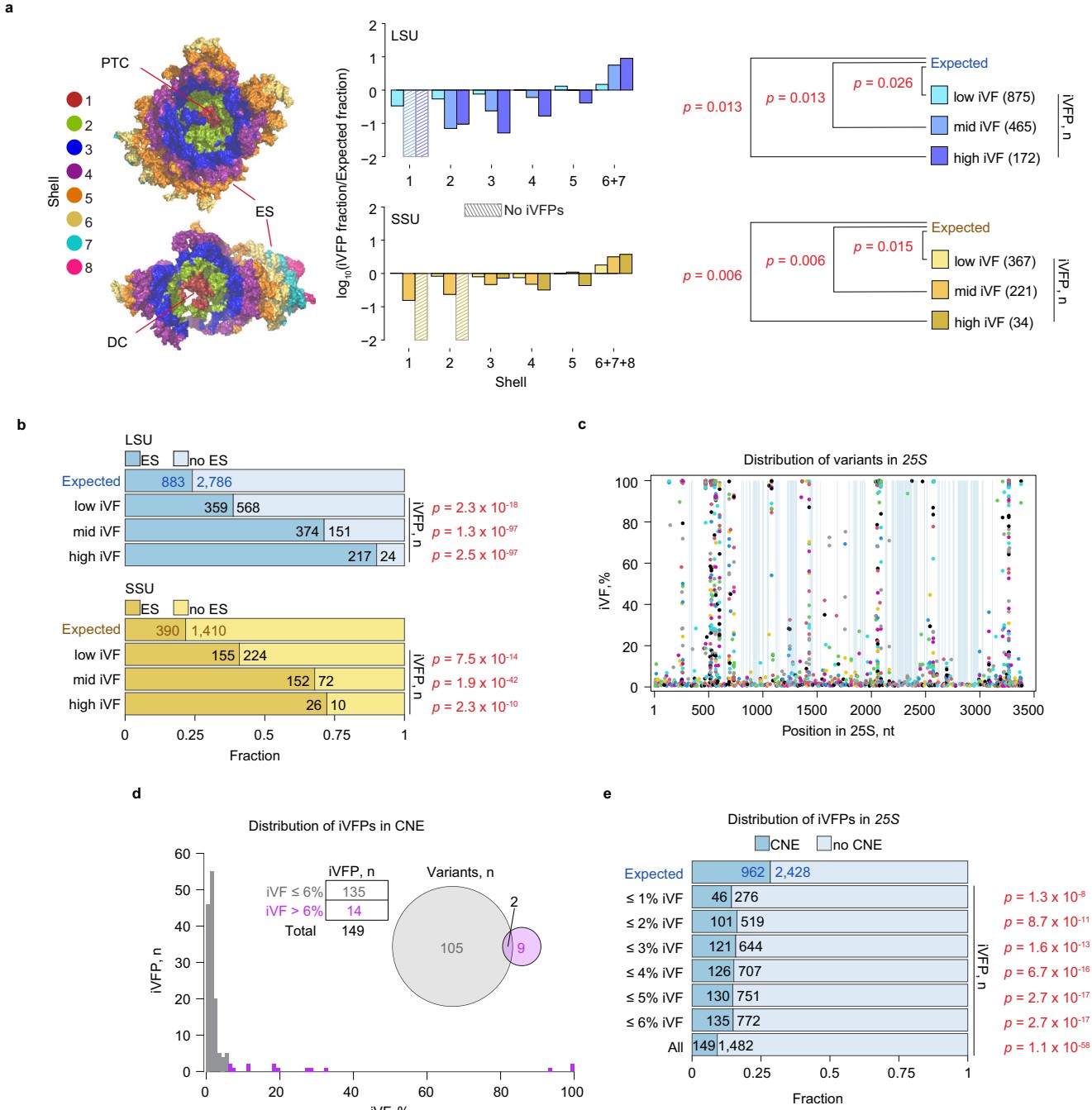

**Fig. 5 | Selection profile of the yeast 80S ribosome. a** Distribution of the number of low, medium, and high iVFPs observed (numbers are indicated in brackets) versus expected in both the LSU and the SSU across their respective concentric shells (see text; depicted to the left). The outermost shells in the LSU and the SSU were pooled together due to the low number of nucleotides. Shells 1 in the LSU and 1 and 2 in the SSU did not have any variants and their log values were artificially set to −2 (hashed bars). Red lines point to the location of: PTC−peptidyltransferase center; DC−decoding center; and ES−expansion segments. **b** Distribution of iVFPs between expansion segments ("ES") and other parts of the rRNA ("no ES"). **c** Distribution of variants across eukaryotic conserved nucleotide elements (CNEs, highlighted in blue). **d** Distribution of iVFPs that are detected in the CNEs. Inset−

number of iVFPs in two groups (below and above 6%; left) and the overlap between the variants from the two groups (right; two variants have iVFPs with iVFs both below and above 6% iVF across different isolates). **e** Distribution of iVFPs with different iVF cutoffs within ("CNE") and outside ("no CNE") of the CNEs. For **a**, $P$-values were calculated with two-sided Wilcoxon rank-sum exact test, for **b** and **e**− with two-sided Fisher's exact test. "Expected" distribution was calculated based on the distribution of total rRNA nucleotides across the shells (**a**) or within and outside of ES (**b**) or CNE (**e**; see "Methods"). All reported $P$-values are adjusted for multiple hypothesis testing using Benjamini−Hochberg correction. Details of statistical tests and source data are provided as a Source data file.

To test whether these variants are expressed, we performed both DNA- and total RNA-sequencing (without poly-A enrichment) of the nine isolates (Supplementary Figs. 25–28; Supplementary Data 6). With the exception of isolate "CFG", we detected variants at very similar iVFs as reported in ref. 32, indicating that iVFs remained largely stable over

the course of culturing. This included the GAC variants, which showed excellent agreement with the data in ref. 32 (Fig. 6e; Pearson's $r = 0.98$, $p = 1.4 \times 10^{-6}$). Importantly, the GAC variants were also detected at matching frequencies in the reads from total RNA-sequencing data (Fig. 6e; Pearson's $r = 0.96$, $p = 7.7 \times 10^{-6}$), indicating that the GAC rRNA

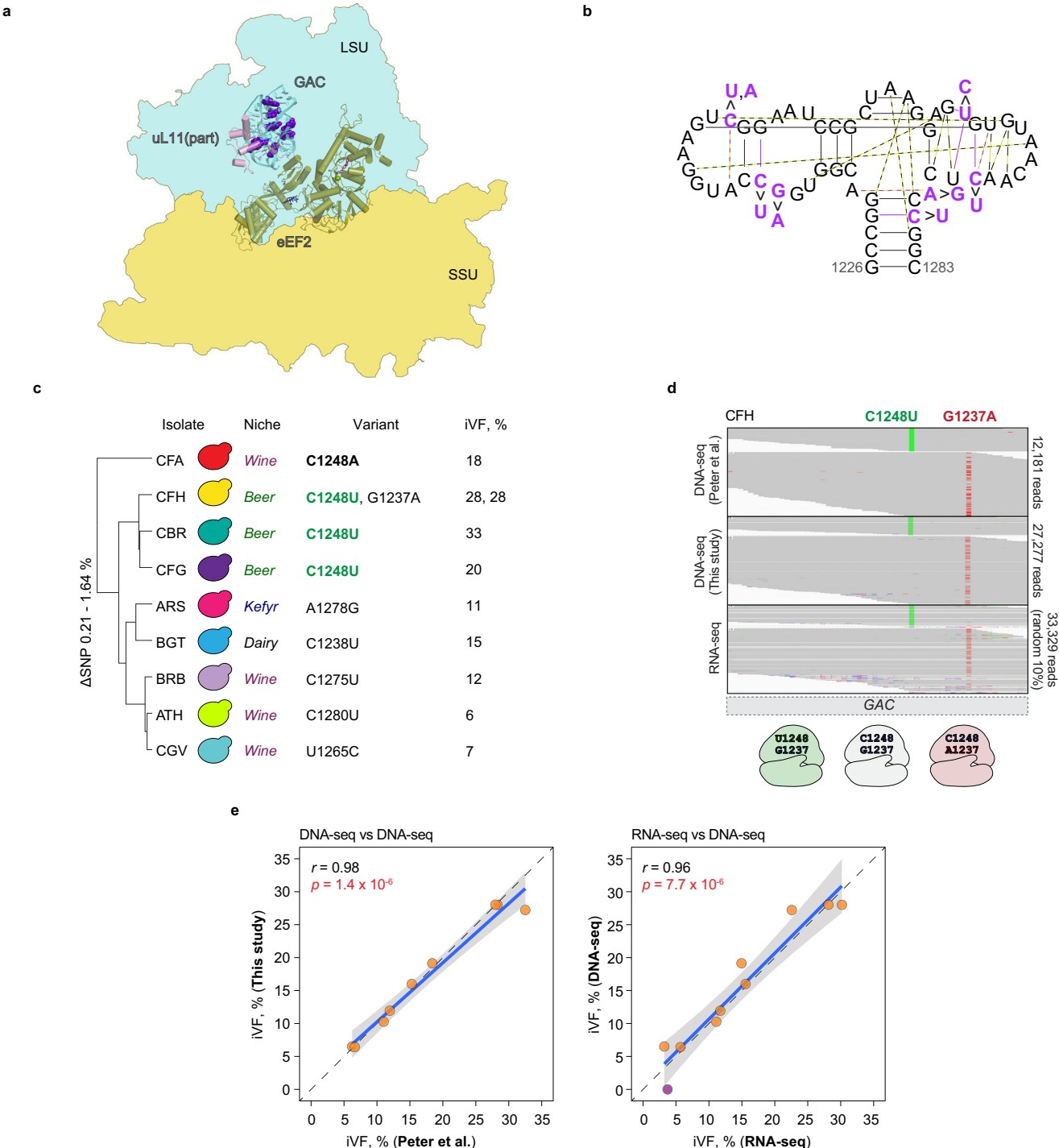

**Fig. 6 | Variability in the GTPase-associated center (GAC). a** The GAC interacts with translational GTPase eEF2 (green) in the 80S yeast ribosome (based on ref. 68; LSU is shown in blue, SSU is shown in yellow). Part of ribosomal protein uL11 is shown in pink. Variant positions are shown in purple with the spherical representation. **b** The secondary structure of the GAC. Variant sites and their interactions are denoted in purple, interactions are shown in lines and weaker (or probable) interactions are shown with dashed lines. **c** Hierarchical clustering of the isolates with SNVs in the GAC (25S nomenclature) based on non-rDNA genome-wide single-nucleotide polymorphism (SNP) differences from ref. 32. Bold highlights a position with more than one variant; green indicates a shared variant. **d** Continuous rRNA and rDNA reads (from this study and ref. 32) of isolate "CFH" (horizontal gray lines) mapped to the GAC. Variant nucleotides are indicated in green and red, with 3 possible variant combinations in the 25S rRNA shown below. Due to the large read coverage of the RNA-seq data (over 300,000 continuous GAC reads), only random

10% of the reads are shown. **e** Correlation between iVFs of the GAC variants detected from DNA-sequencing ("DNA-seq") performed in this study compared to the published DNA-sequencing data from ref. 32, and correlation between iVFs of the variants in the genome ("DNA-seq") and their expression in RNA ("RNA-seq") performed in this study. Isolate "CFG" is excluded because of inconsistent sequencing results between this study and ref. 32. Each dot is a variant; the purple dot represents a variant detected in RNA-sequencing but not in DNA-sequencing data. The diagonal shows theoretical one-to-one correspondence between iVFs of the two experiments. Each dot is a GAC iVFP. The regression line (blue) is plotted with linear regression model. The gray shading is with a 95% confidence interval (CI). Significance test: two-sided Pearson's product-moment correlation; DNA-seq vs DNA-seq: t(7) = 15.0, $p = 1.4 \times 10^{-6}$, $r = 0.98$, 95% CI [0.93, 1]; RNA-seq vs DNA-seq: t(8) = 10.1, $p = 7.7 \times 10^{-6}$, $r = 0.96$, 95% CI [0.85, 0.99]. Source data are provided as a Source data file.

variants are expressed. Moreover, the two variants in isolate "CFH" (C1248U and G1237A) also appeared in mutually exclusive transcripts, indicating that they contribute to different ribosomes.

Secondary structure analysis showed that all but one of the eight variant positions are involved in base-pairing interactions (Fig. 6b). For example, U1248 could create a Watson-Crick pair with A1240, thereby extending the stem of the hairpin, whereas C1275U and C1280U could destabilize stem structures by altering the corresponding Watson-Crick base pairs with G1266 and G1229. Since the GAC undergoes major rearrangements upon GTP hydrolysis[45], these changes could potentially alter the overall geometry of the region and thus affect the interaction with translational GTPases. The relatively high iVFs (6–33%) in such a highly conserved region, together with their ability to alter base pairing, suggests that these variants are either neutral or under positive selection. Interestingly, C1238U, in combination with another variant, was shown to increase translational accuracy in *S. cerevisiae*[46].

To search for additional mid- or high-frequency variants of possible functional importance, we focused on the intersubunit bridges, which link large and small subunits and mediate major rearrangements in the ribosome during translation[47]. We detected 26 variants with mid- and high iVFs within 5 Å of intersubunit bridges (Supplementary Fig. 29; Supplementary Data 5). Among these, variant G1112A in the 18S rRNA (18S rRNA nomenclature; 44.8% iVF in isolate "CDN") is part of helix 27, which dynamically interacts with bridge eB14 during different steps of translation (Supplementary Fig. 30) and is near the decoding center. Intriguingly, the same G1112A mutation was previously shown in mutagenesis studies to decrease stop-codon readthrough in *S. cerevisiae*[48], thus indicating that this naturally occurring variant can alter cell fitness and may be under selection.

## Discussion

Our analyses demonstrate remarkable sequence diversity among rDNA copies both within and among isolates. This diversity is non-uniformly distributed across major rDNA elements (*NTS*, *ETS + ITS*, and *rRNA*), and rDNA variants can persist, and be subject to selection, over a wide range of frequencies. The varying frequencies of rDNA polymorphisms likely arose because of new mutations, random duplication/deletion of rDNA copies, and gene conversion. Moreover, deleterious variants are eliminated, and beneficial variants expanded, as long as their relative copy numbers provide sufficient phenotypic consequences, whereas neutral variants can persist and fluctuate in frequency among rDNA copies for extended periods of evolutionary time. These data suggest a lower rate of concerted evolution than previously thought, and a strong role of directional selection that results in a diverse pattern of intra- and intergenomic rDNA sequence variation within one species. As a result, the substantial within-genome polymorphisms seen in this study and observed across diverse phyla[14–24] imply that selection plays an important role in shaping rDNA diversity across eukaryotes.

The extreme purifying selection against deleterious variants observed here seems at odds with a much higher selection threshold seen in *Drosophila*, where up to 60% of the *28S rRNA* coding sequences are disrupted by R1/R2 retrotransposons[49]. We suggest that these differences can be explained by how the respective mutations contribute to ribosome biogenesis and function. The majority of coding variants observed here are SNVs that are expressed as rRNAs and likely incorporated into ribosomes. Given that translation frequently engages multiple ribosomes on the same transcript, the presence of non-functional ribosomes has the potential to cause partially dominant effects and can thus be sufficiently deleterious to be selected against even at low copy number. Such poisonous effects could explain why defective ribosomes are subject to quality control mechanisms and degraded even if deleterious rRNA variants are expressed at very low levels[50,51]. We expect that variants in *ETS* and *ITS* that yield defective ribosomal particles may have similar effects[52]. By contrast, mutations that disrupt rRNA transcription altogether, as seen in *Drosophila*, are

likely subject to much weaker selection because of the high number of available rDNA copies, which include a substantial number of dormant rRNA genes. Several regulatory mechanisms are known to activate these dormant rRNA genes or generally increase rRNA transcription[53]. These mechanisms are expected to provide a substantial buffer against the inactivation of individual rDNA copies and lead to a higher rDNA copy number threshold for selection. We note that the extreme purifying selection seen in our study may also mask disease-causing variants in the human rDNA, especially since rDNA sequences are largely ignored in genotype-phenotype association studies. Based on our data and previous studies in humans[24], we suggest that considering even low-frequency polymorphisms might be important for investigating the role of rDNA variation in health and disease.

The effects of the detected variants on cell fitness will require further functional analyses. It seems likely that many rRNA nucleotide polymorphisms have the potential to impact cellular physiology and adaptation to the environment. Some variant rRNAs can be differentially expressed in eukaryotes[21,24,54–56], and several rRNA nucleotide changes in yeast confer resistance to antibiotics[57] or affect translational fidelity[46,48]. Additionally, variation among individual rRNA copies can confer benefits in response to stress in E. coli[58]. We uncovered a couple of examples of natural variants that show increased translational fidelity in laboratory experiments[46,48], and may thus be adaptive in certain environments. Our analyses in yeast, along with studies in other organisms[21,24] indicate that rRNA diversity is likely much more pervasive than previously thought. Therefore, sequencing more individuals of a species could reveal substantial hidden variation.

Finally, our results highlight the importance of considering within-species sequence diversity of rDNA in metagenomic analyses, which frequently use short rDNA segments to estimate species numbers and population complexity in environmental or clinical samples[59]. The extensive diversity seen here within a single species and even within single isolates shows that care must be taken when using exact sequence variants or narrowly defined operational taxonomic units for estimating species numbers, as these approaches may lead to substantial overestimates of true species counts.

## Methods

Data were analyzed and visualized using R v.4.0.3 unless otherwise stated. Linear regression models for Figs. 3a, 6e, and Supplementary Figs. 5a, 8, 16, 19, and 20 were calculated using the lm() function in R.

### Read alignment and variant call

Short-read sequencing reads for each isolate (SRA ID: "ERP014555")[32] were used. To remove non-rDNA reads and reduce the rate of false-positive calls downstream, reads of a given isolate were first mapped to an SK1 genome[60] without annotated rDNA using bowtie2[61]: -5 1 -N 0 -p 8. Unmapped reads were mapped once again to the genome with relaxed settings: -5 1 -N 1 -p 8. The -L parameter was set to 22 by default, and the high divergence between nuclear and mitochondrial rDNA allowed the aligner to discriminate between nuclear and mitochondrial rDNA reads. The resulting nuclear rDNA-enriched reads were mapped with relaxed settings to the annotated copy of the S288c rDNA reference sequence (*Saccharomyces* genome database: Chr XII 451575:460711) whose *NTS* sequences were put before the *35S rRNA* instead of after *ETS1* (see "Data availability" for the reference .fsa and .bed files). The parameters for bowtie2 were: -5 1 -N 1 -p 8. The output .sam files were converted into .bam files with SAMtools[62]: samtools view -Sbh -F 12 file.sam | samtools sort -@ 8 -o file.sort.bam -O 'bam' file.bam. Variants were subsequently called using LoFreq[33]. Indel qualities were added to the .sort.bam files with indelqual --dindel, and variants were called afterward (call --call-indels). The output .vcf files were filtered using awk. Variant entries were kept if: (1) nucleotide position >10 and <9100 (to avoid end effects), (2) iVF > 0.005, (3) homopolymer run <4, (4) GC content <0.6 in long indels (>5 nt) in

either reference or alternative sequences, (5) variants were outside of 10-nt stretches of poly A/T/G/C sequences, and 6) the minimal iVF > theoretical one copy iVF computed for each isolate separately (see "rDNA copy number estimation"). In addition, variants with the highest strand bias (≥144; top 12%) were removed to further reduce the rate of false positives[63], and replicates of the same isolate were deduplicated such that only the sample with the highest read coverage was considered downstream. Read coverage was calculated using BEDtools[64]: bedtools genomecov -d -ibam in.bam > coverage.txt. For Fig. 6d and Supplementary Fig. 14b, rDNA reads that fully covered the region of the variants were retrieved from the alignment and visualized in IGV[65]. Random ten percent of GAC RNA reads were sampled using samtools view -b -s 0.1 in.bam > out.bam.

## Pipeline performance
To determine the lowest iVF that can be confidently detected with this pipeline, we performed a series of in silico titrations by "spiking" SK1 reads into S288c reads at different frequencies from 100% to 0.1%. First, paired-end Illumina HiSeq 2500 reads from SK1 and S288c (SRA IDs: "SRR4074258", "SRR4074255") from ref. 60 were mapped using the pipeline, and 10 variants unique to SK1 (not detected in S288c) with iVFs over 95% were tracked (Supplementary Fig. 2). Most of the variants were detected even at the titration point of 0.5%, which corresponds to a variant in as low as 1 rDNA copy (Supplementary Fig. 23). To assess the sensitivity and specificity of the pipeline, we simulated artificial datasets of 56 known SNVs (real data−median: 22 SNVs, range: 7–80 SNVs) and 3 indels (real data−median: 3 indels, range: 1–15 indels) at 0.5% iVF distributed randomly across the rDNA prototype using NEAT-genReads[66] (with python v.2.7), with genomic coverages from 20-fold to 400-fold (3000–60,000-fold for rDNA; real data genome coverage: mean 200-fold) (Supplementary Fig. 2b–d). The sequencing error model (genSeqErrorModel.py) as well as G/C content (computeGC.py) and fragment length (computeFraglen.py) models were inferred from the S288c reads (SRA ID: "SRR4074255"), and the read length (the -R parameter) was set to 150. To better recapitulate the real rDNA coverage (Supplementary Fig. 2b), we modified annotations in the.bed file (see "Data availability"). Reads with "reference" and "alternative" sequences (SNVs and indels) were generated separately and then combined such that the frequency of variants (alternative sequences) was 0.5%. For example, for 3000-fold rDNA coverage reference reads we ran: genReads.py -r rDNA_S288c.fsa -R 150 -o output_file_prefix --bam --pe-model fraglen.p -e seq_error.p --gc-model gcmodel.p -p 1 -M 0 -c 2985 -t rDNA_S288c_benchmark.bed -to 0.4 --rng 456; for alternative reads: genReads.py -r rDNA_S288c.fsa -R 150 -o output_file_prefix --bam --vcf --pe-model fraglen.p -e seq_error.p --gc-model gcmodel.p -p 1 -M 0.015 -c 15 -t rDNA_S288c_benchmark.bed -to 0.4 --rng 123. The variants and error model were the same in all the simulations and the coverage patterns best resembled the real data. The datasets were then processed with the bioinformatics pipeline, and the true known prior variants (generated by NEAT-genReads) were compared to those detected by the pipeline (Supplementary Fig. 2c, d). We did not detect any false-positive calls in the simulated datasets, which is in accordance with the extremely high specificity of the LoFreq caller (false-positive rate <0.2%[33]). Given the actual complexity of the real data, we applied additional filters (see "Read alignment and variant call") to reduce the rate of false-positive variants with likely increased rate of false negative calls.

## rDNA copy number estimation
**Total rDNA copy number.** First, to calculate how many rDNA copies were present in each isolate, we took copy number estimates of *5.8S*, *18S*, and *25S* (*RDN5.8*, *RDN18*, and *RND25*; no reported *RDN5*) reported in the "genesMatrix_CopyNumber.tab" table from ref. 32. Since these sequences are present together in each rDNA copy, we calculated their mean to get an estimate of the rDNA copy number per haploid

genome. We then multiplied these values by the isolates' respective ploidies reported in ref. 32 to get the total number of rDNA copies per isolate. We calculated the number of rDNA copies per clade by taking the median number of rDNA copies of isolates in each clade.

**One-copy rDNA frequency.** For each isolate, we calculated the theoretical one-copy rDNA frequency (i.e., "one copy iVF") based on the total number of rDNA copies (*M*) in each isolate:

$$\text{One copy iVF} = \frac{1}{M} \tag{1}$$

## Sequence entropy
To examine the degree of sequence diversity at each nucleotide position, we calculated Shannon's entropy *H* (Supplementary Data 2) for each rDNA nucleotide position *p*:

$$H(p) = -\sum_{i \in \{A,T,G,C,indel\}} \frac{n_i}{N} \times \ln \frac{n_i}{N} \tag{2}$$

where *N* is the sum of all rDNA copies across the isolates, and $n_i$ is the number of rDNA copies with "A", "T", "G", "C", or "indel" at each position, respectively. We considered total nucleotide composition at each site by pooling all rDNA copies across all isolates (*N* = 193,069) together, and then calculated how many rDNA copies in the pool contained "A", "T", "G", "C", or "indel" at a given position.

## Variant co-occurrence
We estimated pairwise variant co-occurrence by calculating Euclidean distances *D* between iVFs within each isolate:

$$D_{XY} = \sqrt{(X_A - Y_A)^2 + (X_B - Y_B)^2 + \ldots + (X_i - Y_i)^2} \tag{3}$$

where *X* and *Y* are variants; *A*, *B*, … *i* are isolates; $X_A, X_B, \ldots X_i$ are iVFs (from 0.05 to 1, which corresponds to 5–100%). For each possible pair of variants, only isolates that have both *X* and *Y* were considered. Variants *X* and *Y* were considered to occur in the same rDNA copies if: (1) $n \geq 3$ isolates within an ecological niche shared the same pair of variants; (2) $D_{XY} < 0.5$; and (3) the mean of their iVFs

$$\frac{X_A + \ldots + X_i + Y_A + \ldots + Y_i}{n} \tag{4}$$

was <0.05. The parameters were empirically chosen to minimize the difference between variant iVFs and maximize the confidence in the computed variant pairs in the absence of long-read sequencing data, although the confidence would drop for low iVFs. This allowed considering variant pairs that have different iVFs in different isolates but similar iVFs within isolates, e.g., variants *X* and *Y* both have iVF 70% in isolate *A*, iVF 50% in isolate *B* and 7% in isolate *C*, so they always have similar iVFs within an isolate, but different magnitudes in different isolates.

## Correlation between rDNA variants and non-rDNA SNPs
Haploid isolates were excluded from the analysis, and diploid (2n; *N* = 693) and polyploid (3n-5n; *N* = 88) isolates were analyzed separately. Four isolates ("AEC", "AFT", "CEN", and "CFS") were additionally removed from the datasets due to undefined values in ploidy or genome heterozygosity. After that, the number of rDNA variants per isolate (both SNVs and indels) and the respective genome-wide (excluding rDNA) number of heterozygous SNPs per kilobase of heterozygous regions or the value for loss of heterozygosity (from ref. 32) were used to compute Pearson's correlation coefficient. For homozygous diploid isolates, the distribution of rDNA variants was plotted in Fig. 3c.

## rRNA analysis

For the expected distribution of iVFPs in Fig. 5a, the fraction of nucleotides in each shell $F_i$ ($i$ indicates shells 1, 2, 3, 4, 5, and 6+7 for the LSU and 6+7+8 for the SSU) was calculated as the number of nucleotides $n_i$ in a shell (from ref. 38, "Onion" representation) over the total number of nucleotides $N$ in a subunit:

$$F_i = \frac{n_i}{N} \tag{5}$$

Since the number of variants is proportional to the length of a sequence, this suggests that the longer a sequence of interest is, the higher the chance of a variant occurrence in this sequence. We assumed that under a neutral model, the likelihood of an iVFP is higher with increased number of nucleotides in a shell ("expected distribution"). For the LSU ($25S + 5.8S + 5S$), $N = 3428$ instead of the actual 3675 (S288c reference) because 247 nucleotides were missing from the crystal structure and thus could not be used in the shell annotation (since it relies on the 3D structure of the ribosome). For the SSU ($18S$), $N = 1781$ (19 nucleotides were missing from the structure). For the observed distribution of iVFPs, $F_i^*$ was calculated by dividing the number of iVFPs with a chosen range of iVFs (low iVF−[0.5%;5%]; mid iVF−[5%;95%), high iVF−[95%;100%]) in a shell ($m_i$) by the total number of iVFPs ($M$) in the chosen range (low, mid-, or high iVFs):

$$F_i^* = \frac{m_i}{M} \tag{6}$$

Due to the missing segments in the LSU and SSU structures, some iVFPs detected in Fig. 1 were not included in the analysis. The excluded number of iVFPs in the LSU: low iVF−52 out of 927; mid iVF−60 out of 525; high iVF−69 out of 241. The excluded number of iVFPs in the SSU: low iVF−12 out of 379; mid iVF−3 out of 224; high iVF−2 out of 36. We then calculated the ratio between the two values:

$$\log_{10} \frac{F_i^*}{F_i} \tag{7}$$

and used it as the metric for constrained (<0) or relaxed (>0) sequence variability in the ribosome in Fig. 5a, given a significant difference between the expected and observed distributions (Wilcoxon rank-sum test; see text). For example, for low iVFs in shell 1 in the LSU, $F_1 = \frac{82}{3428} = 0.024$, $F_1^* = \frac{7}{875} = 0.008$, and $\log_{10} \frac{F_1^*}{F_1} = -0.48$. For Fig. 5b, Supplementary Fig. 21, and Supplementary Fig. 22, coordinates of ESs and rRNA:ribosomal protein interactions were retrieved from ref. 67. The expected distribution of iVFPs in ESs for Fig. 5b was derived from the number of nucleotides in ESs ($n$; LSU: 883, SSU: 390) versus not in ESs ($N$; LSU: 2786; SSU: 1410)−"ES":

$$\frac{n}{n+N} \tag{8}$$

"no ES":

$$\frac{N}{n+N} \tag{9}$$

For the observed values, iVFPs were counted for each range ($i$ = low, mid-, or high iVFs) within ($m_i$) or outside ($M_i$) of ESs, and the fractions −"ES":

$$\frac{m_i}{m_i + M_i} \tag{10}$$

"no ES":

$$\frac{M_i}{m_i + M_i} \tag{11}$$

were calculated. For Supplementary Fig. 21, the expected distribution of iVFPs in the rRNA alone or rRNA:protein interaction interfaces and the observed values were calculated in the same manner. For Fig. 5c–e, coordinates of eukaryotic-specific conserved nucleotide elements ("CNEs" in this paper) were retrieved from ref. 43. Similarly to Fig. 5b, the expected distribution of iVFPs in CNEs was derived as the fraction of $25S$ nucleotides in CNE ($n = 962$) and in the rest of the $25S$ ($N = 2428$; total $25S$ length−3396)−"CNE":

$$\frac{n}{n+N} \tag{12}$$

"no CNE":

$$\frac{N}{n+N} \tag{13}$$

For the observed values, iVFPs were counted for each range ($i$) indicated in Fig. 5e within ($m_i$) and outside ($M_i$) of CNEs, and the fractions −"CNE":

$$\frac{m_i}{m_i + M_i} \tag{14}$$

"no CNE":

$$\frac{M_i}{m_i + M_i} \tag{15}$$

were calculated.

## DNA sequencing

**Preparation of genomic DNA.** Yeast isolates were grown in 5 ml YPD (20 g/l bactopeptone [ThermoFisher Scientific #211677], 10 g/l yeast extract [ThermoFisher Scientific #212750], 2% glucose [Sigma-Aldrich #G5767]) medium at 30 °C overnight. For each isolate, half of each culture was used to prepare total genomic DNA and the other half was used in total RNA-sequencing (see below). The cells were spun down at 1000 × $g$ for 5 min at 4 °C, and the pellet was washed once in TE buffer (10 mM Tris [VWR #JT4099-2] pH 8.0, 1 mM EDTA [Sigma-Aldrich #E5134]). The pellet was then resuspended in 350 μl of breakage buffer (2% Triton X-100 [Sigma-Aldrich #T8787], 1% SDS [AmericanBio #AB01920-01000], 100 mM NaCl [Sigma-Aldrich #S9888], 10 mM Tris pH 8.0, 1 mM EDTA) with the subsequent addition of 350 μl of phenol:chloroform:isoamyl alcohol (25:24:1, Sigma-Aldrich #P2069) and around 200 mg of glass beads (Sigma-Aldrich #G8772). The cells were lysed using a FastPrep homogenizer (MP Biomedicals) for 2 × 30 s (power = 4.5) at 4 °C and vortexed on an IKA Vibrax VXR for 10 min. The samples were then centrifuged at maximum speed for 15 min at 4 °C, and 200 μl of the upper aqueous layer was transferred to a new tube and incubated with 2 μl RNase A (20-40 mg/ml, Sigma-Aldrich #R4642) at 37 °C for 1 h. The genomic DNA was precipitated with ethanol and resuspended in 25 μl nuclease-free water. The samples were then sonicated using a Bioruptor Pico (Diagenode) for 30 cycles (30 s "on", 30 s "off") to yield 250–600 bp fragments.

**Library preparation and sequencing.** Sequencing libraries were prepared using the TruSeq DNA Sample Prep kit v2-Set A (Illumina #FC-121-2001) for blunting, adding "A" overhangs, and adapter ligation. Excess adapters were removed by using AMPure XP beads (Beckman

Coulter #A63880) and DNA was eluted in 38 μl 10 mM Tris-HCl pH 8.5 (EB from Qiagen #19086). Libraries were amplified with PCR (17 cycles −30 s at 98 °C, 30 s at 60 °C, 30 s at 72 °C) using Phusion polymerase (ThermoFisher Scientific #F530-S) and primers complementary to the adapters. The DNA was cleaned up using the MinElute PCR Purification kit (Qiagen #28006) and eluted with 11 μl EB. Finally, 300–600 bp fragments were purified from 1.5% agarose gel (1xTAE buffer) using the MinElute Gel Extraction kit (Qiagen #28606). Library concentration was determined by Qubit (Invitrogen) using the dsDNA HS Assay Kit (Invitrogen #Q32851), and library quality and size range were determined by TapeStation (Agilent) using High Sensitivity D1000 Screen-Tape (Agilent #5067-5584). Pooled library (18 total−nine from DNA and 9 from total RNA−see below) concentration was determined using a KAPA library quantification kit (Roche #07960140001). Libraries were sequenced on an Illumina NextSeq 500 instrument (2 × 150 bp paired-end reads, HighOutput mode v2.5).

### Total RNA sequencing

Washed cell pellets were prepared as described in "DNA sequencing". Total RNA extraction was performed using the RNeasy MinElute Cleanup kit (Qiagen #74204). First, 600 μl of RLT buffer with 1% β-mercaptoethanol (Sigma-Aldrich #M3148) and 200 mg of glass beads were added to the pellet, and the cells were lysed on an IKA Vibrax VXR for 20 min at 4 °C with subsequent centrifugation at 17,000 × $g$ for 2 min. The supernatant was transferred to a new tube and mixed with an equal volume of 70% ethanol (Pharmco #111000200), and transferred to an RNeasy column to proceed with RNA extraction using the manufacturer's protocol. RNA was eluted from the column with 22 μl of nuclease-free water (Fisher #BP2819). Total RNA was subsequently fragmented using RNA fragmentation reagents (Ambion #AM8740) and cleaned up with the RNeasy MinElute Cleanup kit. RNA was eluted from the column with 9 μl of nuclease-free water.

**First-strand cDNA synthesis.** Random hexamer priming (8 μl total RNA, 1 μl random hexamers (Invitrogen #N8080127), 1 μl 10 mM dNTPs [Promega #U1515]) was used to synthesize complementary DNA. The samples were incubated at 90 °C for 1 min then 65 °C for 5 min, quickly chilled on ice for 1 min, and mixed with 2 μl 10x RT buffer (Invitrogen #18080051), 4 μl 25 mM MgCl₂ (Sigma-Aldrich #M8787), 2 μl 0.1 M DDT (ThermoFisher Scientific #18080093), 1 μl 4U/μl RNase OUT (Invitrogen #10777019), and 1 μl 200U/μl Super-Script III reverse transcriptase (ThermoFisher Scientific #18080093). The samples were incubated at 25 °C for 10 min followed by 50 °C for 50 min. The reaction was heat-inactivated at 75 °C for 15 min. dNTPs were removed with ethanol precipitation by adding 80 μl nuclease-free water, 1 μl glycogen (Ambion #AM9510), 10 μl 3 M sodium acetate (Sigma-Aldrich #S8625) pH 5.2 and 200 μl 100% ethanol followed by incubation at −80 °C for 20 min. The samples were centrifuged at 17,000 × $g$ for 20 min at 4 °C, the supernatant was removed, and the residue was washed with 500 μl cold 75% ethanol. The samples were centrifuged at 17,000 × $g$ for 10 min at 4 °C, the supernatant and all the remaining liquid was removed, and the pellet was air-dried for a few minutes. We then resuspended the pellet in 51 μl nuclease-free water, 1 μl 10x RT buffer, 1 μl 0.1 M DTT, and 2 μl 25 mM MgCl₂.

**Second-strand cDNA synthesis.** We further added 15 μl 5X second-strand buffer (100 mM Tris-HCl pH 6.9, 450 mM KCl [Sigma-Aldrich #P3911], 23 mM MgCl₂ [Sigma-Aldrich # M2670], 0.75 mM β-NAD⁺ [NEB #B9007S], 50 mM (NH₄)₂SO₄ [Fisher #BP212R-1]), 2 μl 10 mM dA/G/C/UTPs (Promega #U1335), 0.5 μl 1U/μl *E. coli* DNA ligase (NEB #M0205L), 2 μl 10U/μl DNA Pol I (Enzymatics #P705L), and 0.5 μl 5U/μl RNase H (Enzymatics #Y922L). The mixture was incubated at 16 °C for 2 h and the resulting double-stranded DNA was purified using the MinElute PCR Purification kit with 20 μl EB for elution. After that, the

libraries were prepared and sequenced using the same protocol as described in "DNA sequencing".

### Structural analysis

Ribosome structures PDB IDs: "4v88" (assembly 1)[67] and "6woo"[68] were visualized and analyzed in PyMOL v.2.4.1 (using python v.3.7.9). Since ribosomal parts undergo major rearrangements during translation[47,69], we considered variants that are directly in or within 5 Å of the inter-subunit bridges as potential candidates located in the functionally important regions. Coordinates of the intersubunit bridges were extracted from ref. 67. Analysis in Supplementary Fig. 30 was made with local structural alignment of the region (18S:1110-1136 together with eL41) from "4v88" and "6woo" in PyMOL. For Fig. 6b and Supplementary Fig. 14c, the NDB (http://ndbserver.rutgers.edu/)[70] was used to retrieve nucleotide interactions in the 25S rRNA (PDB ID: "4v88").

### Reporting summary

Further information on research design is available in the Nature Portfolio Reporting Summary linked to this article.

## Data availability

Sequence data generated in this study have been deposited in the SRA database under accession code "PRJNA867718". The data generated in this study are provided in the Supplementary Information files. Supplementary Data 1 is a tab-delimited table of filtered rDNA variants with their intragenomic frequencies. Supplementary Data 2 is a tab-delimited table of calculated Shannon's entropy values at each position of the rDNA. Supplementary Data 3 is a tab-delimited table of variant pairs with consistently similar frequencies across all isolates. Supplementary Data 4 is a tab-delimited table of variant pairs with consistently similar frequencies calculated by niche. Supplementary Data 5 is a tab-delimited table with annotations of variants in rRNA genes. Supplementary Data 6 contains (1) raw_rDNA_var_calls−raw rDNA.vcf files prior to additional filtering; (2) Sequencing_Sultanov_et-al−raw rDNA.vcf files (for DNA- and total RNA sequencing) generated in this study and the rDNA coverage for each sample; (3) rDNA_S288c.fsa−the S288c rDNA copy prototype sequence used in this study; (4) rDNA_S288c.bed−annotations associated with the rDNA prototype; (5) rDNA_S288c_benchmark.bed−annotations for bench-marking; (6) positions_in_homopolymers.txt−nucleotide positions in the S288c rDNA prototype that are embedded in the 10-nt poly(A/T/G/C) sequences. Other previously published sequencing data used in this study are available in the SRA database under accession codes "ERP014555", "SRR4074258", and "SRR4074255"). Previously published structures used in this study are available in the PDB database under accession codes "4v88" and "6woo". Source data are provided with this paper.

## Code availability

The computer scripts used for processing and analyzing rDNA sequencing reads are available at https://github.com/hochwagenlab/rDNA[71].

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

## Acknowledgements

We thank Matthew Rockman, David Gresham, Mark Siegal, Loren Williams, David Fitch for helpful discussions, and Joseph Schacherer for sharing strains. We also thank Genomics Core at NYU CGSB for technical assistance, and NYU IT High Performance Computing for their resources. This work was supported by the National Institutes of Health (R01GM111715 and R01GM123035 to A.H.).

## Author contributions

Conceptualization: D.S. and A.H.; investigation and formal analysis: D.S. and A.H.; coding: D.S.; writing—original draft, review & editing: D.S. and A.H.

## Competing interests

The authors declare no competing interests.
