## [Peer Review File · Nature Communications]

Varying strength of selection contributes to the intragenomic diversity of rRNA genesReviewers' Comments:

Reviewer #1:

Remarks to the Author:

In this article the authors have investigated the level of sequence heterogeneity across ribosomal DNA (rDNA) sequence elements (ETSs, ITSs, 5S, 5.8S, 18S, 25S, NTSs) by analysing the Illumina reads from 918 *Saccharomyces cerevisiae* isolates of the 1002 Yeast Genomes Project (Peter et al. 2018). The authors claim their data reveal a **pervasive** heterogeneity in rDNA sequences **within** and **between** genomes (lanes 124-125) and state as soon as from the abstract that these results challenge the model of concerted evolution, which is thought to mould the rDNA highly repeated sequences so that repeats are maintained in the genome with very similar sequences but differ between genomes/within species. The authors also state they show that rDNA copies in yeast are far from homogenous, both among and **within** isolates (abstract lanes 11-12). Despite the amount of analyses presented I am not convinced the work supports authors' conclusions and claims especially about challenging the concerted evolution model (or any evolutionary model) and the short discussion paragraph dedicated to that (lanes 237-247) does not convince me either.

While rDNA deep sequencing data from 918 *S. cerevisiae* isolates offer the opportunity to capture sequence variation **within** and **between** genomes, there are only two examples of intragenomic variation presented (figure 5 and supplementary figure 11) probably because the heterogeneity within genomes is not as pervasive as claimed. Most of the work is based on analysing all together the reads from the 918 *S. cerevisiae* isolates or the reads within each ecological niche or clade (without explanation why ecological niches or clades are considered from one figure to another). The authors explain their methodology and how they calculate a 'variant frequency' (VF) and a 'variant frequency polymorphism' and then justify that their approach effectively allows to consider a VF as low as 0.4% because it allows detection of variants that are present in only one or few rDNA copies within a genome (lanes 66-67). By the way a VF at 0.5% is mentioned in the Methods (not 0,4%) as well as for some analyses (without explanation) while 1% is used in the abstract (lane 17).

To justify using a 0.4% VF, they need to state that the median rDNA copy number among isolates lies around 180 copies (lane 65-66), but there is no explanation how they reach that number and if that number is **per haploid genome or not** and the Methods part does not help since it is only two lanes there (lanes 334-335), in which the **calculation of a mean** copy number is mentioned. It happens that the mean rDNA copy number for those wild *S. cerevisiae* isolates is ~ 90 per haploid genome with a very variable range among isolates of 20 to 230 copies per haploid genome (Peter et al., Nature 2018; West et al. Systematic Bio 2014; Liti et al. Nature 2009). With a mean copy number at 90 per haploid genome and a VF at 0.4%, a variant will then be considered when present in 0.36 rDNA copy only, which is rather questionable. Moreover, there is no explanation about how each *S. cerevisiae* isolate rDNA copy number has been (or not) taken into account in the analyses, consequently the presence of variants in the rDNA units of isolates with low rDNA copy number is dubious while seemingly used to assess the heterogeneity in the rDNA sequences. Since the authors write 'the diversity landscape across rDNA shows clear functional stratification, suggesting that different copy-number thresholds for selection shape rDNA diversity' (lanes 14-16), the reader would have expected that the large variability of rDNA copy-number among isolates would have been considered in the analyses and discussed. Another striking point is claiming that 'the A817C, T2227C and A8934G (variants) evolved as early as in the Chinese lineage, which dates back around 15,000 year ago' (lanes 153-154) when all three variants display an obvious very low VF (Figure 3) in one of the only two isolates that Chinese III clade contains. By the way, the exact VF value for those variants cannot be found in Supplementary file 1 since therein isolates are listed under an unexplained 3 letter code (used for some isolates in Figure 5b as well). Lanes 43-44 'Here, we took advantage of the high sequencing coverage of the 1002 Yeast Genomes Project to **quantitatively assess** within- and between-genome heterogeneity in the rDNA in *Saccharomyces cerevisiae*', when a quantitative assessment is **key**, the reader should not expect to read things

as 'a substantial number of...' (lane 93-94), or 'we also observed numerous variant pairs...' (lane 116) without any quantification presented.

Since 2014, a new system for naming ribosomal proteins is used/recommended by the ribosome community to eliminate confusion between identical names to ribosomal proteins from different species (Ban, N., Beckmann, R., Cate, J.H.D., Dinman, J.D., Dragon, F., Ellis, S.R., et al. (2014) A new system for naming ribosomal proteins. *Curr Opin Struct Biol* 24: 165–169). It would have been useful to have that naming used in the text to help evaluating the relevance of the potentially very interesting data presented in figure 5 and in supplementary figures 16 and 17.

In conclusion, I consider there are too many flaws and weakly elaborated points in that work to recommend its publication.

Reviewer #2:

Remarks to the Author:

This is an excellent executed study and written paper that reveal novel insights into the rDNA array evolution in the budding yeast *S. cerevisiae*. Both text and figures are of high level. The study leave the open question on whether the rDNA heterogeneity consequence can lead to phenotypic / fitness variation. Given the same sequenced collection has been broadly phenotype for many traits (e.g. some here <https://www.nature.com/articles/s41559-022-01671-9>), can this be really explored by testing association for both heterogeneity and copy number with the various fitness components?

Reviewer #3:

Remarks to the Author:

Sequence heterogeneity investigations of ribosomal DNA (rDNA) have recently been made that indicate evolutionary conservation and physiological significance. Recent advances in sequencing technologies further enable such investigations. Strong signals of purifying selection in genomic data have, however, yet to elucidated. Here, the authors hypothesize there are inter- and intra-genomic variations that determine signatures of purifying selection. The authors utilize sequences from the 1002 Yeast Genomes Project to assess this heterogeneity.

The authors first looked at the genomes of 918 yeast sequences obtained from a plethora of environments. Using a very conservative variant caller, the authors determined a large cohort of variants and their distributions across the rDNA loci. In addition, they found differences in the rRNA-encoding regions versus others, indicating substantially less variation in rRNA. They further delved into variant pairs and catalogued variants that occurred in a pairwise fashion.

Next, the authors looked at the persistence of variants in the yeast's genotype. They achieved this by comparing the rDNA variant frequencies to the rest of the genome, as well as to homozygosity in yeast isolates. Further, they analyzed a phylogenetic tree to look at VF persistence.

The authors next looked at the location of where these variants may exist in the ribosome, stratifying in a concentric way the locations of the variants from the outermost portions of the large/small ribosomal subunits to their innermost portions. They found that there seems to be the fewest number of variants in highly functional locations of rRNA, while the variant frequency increases as you move towards the outer shell of the ribosome. The authors also claim that the most conserved elements of 25s seem to also not have a high VF. These findings are consistent with those of Parks et al. *Science Advances* 2018, who performed similar analyses for human and mouse.

Lastly, the authors look at how variants affect functional centers of the ribosome, including the GTPase association center and intersubunit bridges.

On the whole, the paper was well-researched and well-written. The overall crux of this paper suggests that there does exist rDNA heterogeneity that is under selective pressures. Similar conclusions have been made for human/mouse (Parks et al. *Science Advances* 2018; Kurylo et al. *Cell Reports* 2018) will no doubt further open the field to research questions on ribosomal sequences and how they differ.

As I see it, this manuscript adds an important new set of insights to this emerging area of study by indicating supporting evidence for “purifying” selection of conserved variants in the assembled yeast ribosome. A journal of Nature Communications impact and import appears to be a suitable choice for publication pending the following recommended improvements and clarifications:

Questions:

1. Have the authors made any attempt to measure expression of variant rDNA alleles in yeast and their incorporation into actively translating ribosomes? This seems critical to validating that the variants identified actually occur in transcribed regions of the genome that give rise to functional ribosomes rather than rDNA-like regions of the genome.
2. There is not a lot of discussion as to how the environment of the yeast plays a role on purifying selection. While Fig 1c discusses it briefly, the authors do not consider those VFs that exist across multiple species that may serve to differentiate how the yeast has grown in a certain environment. On page 6 lines 2-3, the authors state, “Almost half (42%) of those variants were shared across multiple isolates (Supplementary Fig. 5) but occurred at various VFs (i.e. in different proportions of rDNA copies) and thus represent variant frequency polymorphisms (VFPs; Fig. 1a).” But the authors do not state how these VFPs are distributed across the different niches. A further analysis as to the role the environment plays should be pursued. Perhaps a PCA plot colored by niche may illuminate this point.
3. The authors make a clear distinction between VFs and VFPs, but it is not clear as to which variants are more polymorphic than others. For example, Fig S7 describes the number of shared isolates per variant, but they do not describe how polymorphic each of those sites are. How much sequence entropy/information content exists at each site?
4. On page 7, line 21, where the authors discuss variant pairs, several questions arise:
 - a. Based on Supplementary Fig. 10, it seems like the distances computed were done on a niche basis but were not done on the overall dataset. Analyses on all isolates should be reported as well.
 - b. For the methods used in this paragraph, some of the conditions seemed to be arbitrarily chosen. Why does “Dxy” have to be ≤ 0.5 ? Why does “n” have to be ≥ 3 ?
 - c. The overarching point of this paragraph, which appears to be that variants occur in pairs, should be made more obvious.
5. In the sections titled Selection profile of the yeast ribosome and Variants in functional regions, the authors describe functional effects of the variants in the ribosome and how they may alter the geometry of the region, e.g. the GAC or the locations of such variants. The authors state that in the innermost shell, we see extremely low VFs. Yet when one peruses Supplementary File 3, one can see various variants that have high VF. A brief discussion about these positions of variance and how they may affect the decoding center or the peptidyltransferase center seems quite important and clarifications seem warranted. It may help further elucidate which specific nucleotides have indispensable connections.
6. The section rRNA analysis in the methods requires clarification:
 - a. Why is the expected distribution of VFPs defined to just be the fraction of nucleotides in each shell? This would appear to suggest that under a neutral selection model, 1 VFP per nucleotide is expected. I do not see how this is justified.
 - b. The chosen ranges for the VFs seem arbitrary. Why is a low VF from [0.4%-5%]? Why not [0.4%-10%]?
7. Lastly, there has been some work on how sequence differences in rDNA loci have conferred fitness benefits to organisms (Kurylo et al Cell Reports 2018). The authors discuss some functional changes that occur with variants, but do not address questions of fitness. A paragraph or two discussing how some of these changes might confer a fitness advantage would be engaging.

Additional questions/suggestions:

1. Next time, please set your margins in the “justify” setting in Microsoft Word for ease of viewing.
2. On page 6, line 11 the authors report the mean number of isolates per variant, then proceed to compare the distributions via a non-parameterized test. For consistency, please report the medians instead.
3. On page 6, lines 13-16, the authors state a possible reason as to why rRNA genes have fewer isolates per variant than their ETS+ITS and NTS counterparts. Please provide a reference showing this or move this statement to the discussion.

4. On page 6, line 22, the authors state, "...indicating regions of increased permissiveness for variation." What does "permissiveness" mean in this context?
5. Please fix your usage of i.e. I.e. stands for id est which means, "that is." E.g. stands for exempli grata which means, "for example."
6. On page 9, lines 17-19, please cite a reference or put in the discussion.
7. On page 9, line 16, non-significant P-values do not need to be reported.
8. On pages 9-10, lines 23 and 1-2, please clarify the statement, "In general, the same sequence changes presented repeatedly within clades and across related clades, while other possible changes of the same nucleotides were largely undetectable."
9. On page 11, line 16, why did you not do a p-value correction here as you did previously?

We thank the editor and the reviewers for their overall enthusiasm for this work. We believe some of the comments regarding the approach and data analysis stemmed from our brief explanations as well as terminology that could lead to misunderstandings of some of the results. To improve clarity, we expanded the Introduction and Discussion, and re-wrote the Results sections. We also re-analyzed our data based on the reviewers' suggestions, and all of the main conclusions remained as stated.

REVIEWER COMMENTS

Reviewer #1 (Remarks to the Author):

In this article the authors have investigated the level of sequence heterogeneity across ribosomal DNA (rDNA) sequence elements (ETSs, ITSs, 5S, 5.8S, 18S, 25S, NTSs) by analysing the Illumina reads from 918 *Saccharomyces cerevisiae* isolates of the 1002 Yeast Genomes Project (Peter et al. 2018). The authors claim their *data reveal a pervasive heterogeneity in rDNA sequences within and between genomes* (lanes 124-125) and state as soon as from the abstract that these results challenge the model of concerted evolution, which is thought to mould the rDNA highly repeated sequences so that repeats are maintained in the genome with very similar sequences but differ between genomes/within species. The authors also state they *show that rDNA copies in yeast are far from homogenous, both among and within isolates* (abstract lanes 11-12). Despite the amount of analyses presented I am not convinced the work supports authors' conclusions and claims especially about challenging the concerted evolution model (or any evolutionary model) and the short discussion paragraph dedicated to that (lanes 237-247) does not convince me either.

We thank the reviewer for their careful reading of the manuscript. As outlined in detail below, our analyses are based on the rDNA sequences from individual genomes. This allowed us to show that variants exist both within and between genomes. We expanded the Introduction and Discussion, and re-wrote the Results sections to improve clarity and better describe the reasoning underlying our conclusions.

While rDNA deep sequencing data from 918 *S. cerevisiae* isolates offer the opportunity to capture sequence variation **within** and **between** genomes, there are only two examples of intragenomic variation presented (figure 5 and supplementary figure 11) probably because the heterogeneity within genomes is not as pervasive as claimed.

We disagree with this comment but believe our new description of VFs and VFPs (now "iVF" and "iVFP") as well as additional figures in the main and supplemental text will help clarify this point. Each dot in **Fig. 1c** is a variant **within** a genome. The magnitude of each dot is its **intragenomic** variant frequency (now called "intragenomic variant frequency", or iVF) and represents the frequency of this variant within a genome of a particular isolate. This variant frequency does not represent population-wide genomic frequency, and only refers to the frequency of this variant among the rDNA copies of that isolate's genome.

Additionally, a plot added in the revised manuscript (**Fig. 1d**, also below) shows that the median number of *non-fixed* intragenomic rDNA variants (i.e. at least 2 rDNA sequences exist at a given position in the rDNA of an isolate) is 13 per isolate genome (not in the population), which further describes the intragenomic rDNA variation.

The *between* genome variation of the isolates is also apparent from the plot in **Fig. 1c** because variants are differentially colored based on isolates. The plot shows that variants can be observed in one or a subset of isolates but not in other isolates. This distribution is quantified in **Supplementary Fig. 5b**, which shows that over 50% of variants are uniquely present in 1 isolate, and ~80% present in <5 isolates. These analyses demonstrate pervasive between-genome variation of rDNA sequences.

Most of the work is based on analysing all together the reads from the 918 *S. cerevisiae* isolates or the reads within each ecological niche or clade (without explanation why ecological niches or clades are considered from one figure to another).

To maximize statistical power, the majority of our analyses were performed on the entire collection of variants in the dataset. We included individual analyses of clades or ecological niches only at points where the patterns seen in individual clades or niches clearly deviated from the general pattern.

The authors explain their methodology and how they calculate a 'variant frequency' (VF) and a 'variant frequency polymorphism' and then justify that their approach *effectively allows to consider a VF as low as 0.4% because it allows detection of variants that are*

present in only one or few rDNA copies within a genome (lanes 66-67). By the way a VF at 0.5% is mentioned in the Methods (not 0,4%) as well as for some analyses (without explanation) while 1% is used in the abstract (lane 17).

We now used 0.5% throughout the revised manuscript. We initially used 0.4% because we noticed from our benchmarking analysis (**Supplementary Fig. 2**) that the variant caller called variants with true iVF = 0.5% as variants with iVFs from 0.4% to ~0.7% (in high coverage data). The 1% value used in the abstract referred not to the lowest detectable iVFs but to the level of stringent purifying selection from **Fig. 5e** (“*Selection profile of the yeast ribosome*”). However, we now revised the abstract to say “in only a small fraction of rRNA gene copies”.

To justify using a 0.4% VF, they need to state that *the median rDNA copy number among isolates lies around 180 copies* (lane 65-66), but there is no explanation how they reach that number and if that number is **per haploid genome or not** and the Methods part does not help since it is only two lanes there (lanes 334-335), in which the **calculation of a mean** copy number is mentioned.

We now clarified that the median rDNA copy number refers to the total number of rDNA copies per genome. In addition, we added **Fig. 1b**, showing the rDNA copy number distributions (both haploid and total) in each clade, and elaborated on our rDNA copy number calculation in the Methods section.

To calculate how many rDNA copies were present in each isolate, we took the copy number estimates of 5.8S, 18S and 25S (*RDN5.8*, *RDN18* and *RND25*; *RDN5* was not available) reported in *Peter et al., 2018* in their “genesMatrix_CopyNumber.tab” table. Since all 3 rRNAs are present together in each rDNA copy, we calculated their mean copy number to get an estimate of the rDNA copy number per haploid genome. We then multiplied these values by the isolates’ respective ploidies reported in (*Peter et al., 2018*) to get the total number of rDNA copies per isolate. We calculated rDNA copy number per clade by taking the median of rDNA copy number of isolates in each clade.

It happens that the mean rDNA copy number for those wild *S. cerevisiae* isolates is ~ 90 per haploid genome with a very variable range among isolates of 20 to 230 copies per haploid genome (Peter et al., Nature 2018; West et al. Systematic Bio 2014; Liti et al. Nature 2009). With a mean copy number at 90 per haploid genome and a VF at 0.4%, a variant will then be considered when present in 0.36 rDNA copy only, which is rather questionable. Moreover, there is no explanation about how each *S. cerevisiae* isolate rDNA copy number has been (or not) taken into account in the analyses, consequently the presence of variants in the rDNA units of isolates with low rDNA copy number is dubious while seemingly used to assess the heterogeneity in the rDNA sequences.

We agree that there is a lot of variation in rDNA copy numbers, and therefore we now 1) increased the threshold of the lowest iVF to 0.5% (this is from one of the comments above); and 2) for each isolate, we calculated the theoretical one-copy rDNA frequency (“one_copy_VF”) based on the total number of rDNA copies in each isolate (see Methods section and **Supplementary File 1**) and only considered variants with iVFs \geq one_copy_VF.

For each isolate *individually*:

One_copy_VF = $1/(\text{total_number_of_rDNA copies in this isolate})$.

Finally, we also removed variants that occur in homopolymer regions (A/T/G/C)₁₀ since these regions are especially prone to having high rates of false-positives due to polymerase slippage and incorrect base calls.

Since the authors write ‘*the diversity landscape across rDNA shows clear functional stratification, suggesting that different copy-number thresholds for selection shape rDNA diversity* (lanes 14-16), the reader would have expected that the large variability of rDNA copy-number among isolates would have been considered in the analyses and discussed.

rDNA copy-number is indeed variable among isolates, hence we used iVFs which reflect the fraction of variant rDNA copies within an isolates rather than the absolute copy number values, and acts as internal normalization for the rDNA copy number in each isolate. We consider rDNA copy number variation in **Fig. 2**, **Supplementary Fig. 5a** and **Supplementary Fig. 17** that serves as a control to **Fig. 3**. Finally, to further elaborate on the variability of rDNA copy number, we added and discussed **Fig. 1b**, which shows rDNA copy numbers across clades.

Another striking point is claiming that *the A817C, T2227C and A8934G (variants) evolved as early as in the Chinese lineage, which dates back around 15,000 year ago* (lanes 153-154) when all three variants display an obvious very low VF (Figure 3) in one of the only two isolates that Chinese III clade contains.

We agree that the Chinese clades indeed contain very few isolates and now relaxed our statement by instead using variants in Wine clades in our estimation of rDNA variant persistence. Wine *S. cerevisiae* domestication times from Peter et al., 2018

[Supplementary Note 4] are 1,411 and 1,555 years based on two different estimates of spontaneous mutation rates. Additionally, we changed the figure by including all the clades and iVFs as they are reported in **Supplementary File 1**.

By the way, the exact VF value for those variants cannot be found in Supplementary file 1 since therein isolates are listed under an unexplained 3 letter code (used for some isolates in Figure 5b as well).

The three-letter codes are the standardized names used in the *Peter et al., 2018* study (now clarified in the manuscript).

Lanes 43-44 '*Here, we took advantage of the high sequencing coverage of the 1002 Yeast Genomes Project to **quantitatively assess** within- and between-genome heterogeneity in the rDNA in *Saccharomyces cerevisiae**', when a quantitative assessment is **key**, the reader should not expect to read things as 'a substantial number of...' (lane 93-94), or 'we also observed numerous variant pairs...' (lane 116) without any quantification presented.

We now added the numbers in the text as well.

Since 2014, a new system for naming ribosomal proteins is used/recommended by the ribosome community to eliminate confusion between identical names to ribosomal proteins from different species (Ban, N., Beckmann, R., Cate, J.H.D., Dinman, J.D., Dragon, F., Ellis, S.R., et al. (2014) A new system for naming ribosomal proteins. *Curr Opin Struct Biol* 24: 165–169). It would have been useful to have that naming used in the text to help evaluating the relevance of the potentially very interesting data presented in figure 5 and in supplementary figures 16 and 17.

Thank you for highlighting this issue. The names are now changed throughout the manuscript.

In conclusion, I consider there are too many flaws and weakly elaborated points in that work to recommend its publication.

We hope our revision clarifies this reviewer's concerns.

Reviewer #2 (Remarks to the Author):

This is an excellent executed study and written paper that reveal novel insights into the rDNA array evolution in the budding yeast *S. cerevisiae*. Both text and figures are of high level. The study leave the open question on whether the rDNA heterogeneity consequence can lead to phenotypic / fitness variation.

We thank this reviewer for their positive evaluation of our work.

Given the same sequenced collection has been broadly phenotype for many traits (e.g. some here <https://www.nature.com/articles/s41559-022-01671-9>), can this be really explored by testing association for both heterogeneity and copy number with the various fitness components?

This is indeed something we have wanted to explore in the future since it would require additional analyses and controlled experiments with *in vivo* validation, since: 1) variants can co-occur and affect the phenotype through epistasis, especially in the rRNA genes, where two variants located in a stem of a stem-loop can re-form the bonds and do not affect the overall structure and function of the ribosome, whereas individually they can locally melt the stem and non-neutrally affect the phenotype. Although our co-occurrence analysis gives some overview of which variants might occur in a pairwise fashion, ultimately it needs to be validated by long-read sequencing data (unless they occur close to each other within the range of a short read); and 2) rDNA variants can co-evolve with mutations in other genes/proteins and required a genome-wide analysis of the data.

However, with the current dataset we were able to test several assumptions in particular:

- 1) We tested an association between the most prevalent 18S and 25S variants and stress conditions used to phenotype isolates in *Peter et al., 2018* (**Supplementary Fig. 19, 20**) since the number of the variants provides enough power for the analysis. For the tested conditions, there was absent or only weak association between growth phenotype and iVFs of either of the two variants.

- 2) We stratified variants and rDNA copy number based on the domestication status of each isolate from *De Chiara et al., 2022* (**Fig. 2**) and found a significant increase of the number of rRNA gene variants in the “domesticated” vs “wild” groups, and observed significantly lower rDNA copy number in “human” vs other groups.

Reviewer #3 (Remarks to the Author):

Sequence heterogeneity investigations of ribosomal DNA (rDNA) have recently been made that indicate evolutionary conservation and physiological significance. Recent advances in sequencing technologies further enable such investigations. Strong signals of purifying selection in genomic data have, however, yet to be elucidated. Here, the authors hypothesize there are inter- and intra-genomic variations that determine signatures of purifying selection. The authors utilize sequences from the 1002 Yeast Genomes Project to assess this heterogeneity.

The authors first looked at the genomes of 918 yeast sequences obtained from a plethora of environments. Using a very conservative variant caller, the authors determined a large cohort of variants and their distributions across the rDNA loci. In addition, they found differences in the rRNA-encoding regions versus others, indicating

substantially less variation in rRNA. They further delved into variant pairs and catalogued variants that occurred in a pairwise fashion.

Next, the authors looked at the persistence of variants in the yeast's genotype. They achieved this by comparing the rDNA variant frequencies to the rest of the genome, as well as to homozygosity in yeast isolates. Further, they analyzed a phylogenetic tree to look at VF persistence.

The authors next looked at the location of where these variants may exist in the ribosome, stratifying in a concentric way the locations of the variants from the outermost portions of the large/small ribosomal subunits to their innermost portions. They found that there seems to be the fewest number of variants in highly functional locations of rRNA, while the variant frequency increases as you move towards the outer shell of the ribosome. The authors also claim that the most conserved elements of 25s seem to also not have a high VF. These findings are consistent with those of Parks et al. *Science Advances* 2018, who performed similar analyses for human and mouse.

Lastly, the authors look at how variants affect functional centers of the ribosome, including the GTPase association center and intersubunit bridges. On the whole, the paper was well-researched and well-written. The overall crux of this paper suggests that there does exist rDNA heterogeneity that is under selective pressures. Similar conclusions have been made for human/mouse (Parks et al. *Science Advances* 2018; Kurylo et al. *Cell Reports* 2018) will no doubt further open the field to research questions on ribosomal sequences and how they differ. As I see it, this manuscript adds an important new set of insights to this emerging area of study by indicating supporting evidence for “purifying” selection of conserved variants in the assembled yeast ribosome. A journal of *Nature Communications* impact and import appears to be a suitable choice for publication pending the following recommended improvements and clarifications:

We thank this reviewer for their critical and helpful evaluation of our study and for seeing the potential of our work for the field.

Questions:

1. Have the authors made any attempt to measure expression of variant rDNA alleles in yeast and their incorporation into actively translating ribosomes? This seems critical to validating that the variants identified actually occur in transcribed regions of the genome that give rise to functional ribosomes rather than rDNA-like regions of the genome.

This is a great point and we now performed and added results of DNA- and RNA-sequencing experiments on the isolates with variant GAC sequences (**Fig. 6d, Supplementary Fig. 25-28**). We observe an excellent agreement between our DNA-seq data and those of *Peter et al., 2018*, indicating that these variant frequencies indeed represent intragenomic variation (rather than an artifact from yeast colony picking) and that they are stable over time. We also found excellent agreement between variant's

iVFs found in the GAC from DNA-seq and total RNA-seq data. We note here and in the text that the signal from RNA-seq data is indeed coming from the ribosomal RNA rather than rDNA-like regions because:

- 1) *S. cerevisiae* genome is very compact (genes are very close to each other and predominantly intronless) and the yeast genome project observed no rDNA pseudogenes, in contrast to much larger genomes like in humans, where a portion of pseudogenes indeed comes from rDNA (10.1093/gbe/evw307);
- 2) The very high coverage of the RNA-seq data (**Supplementary Fig. 25**) suggests the reads are indeed coming from ribosomal RNA rather than randomly transcribed DNA regions, since rRNA is the most abundant RNA in the cell.

Therefore, our results suggest these transcripts are indeed ribosomal RNA that is transcribed from rDNA and presumably incorporated into ribosomes due to the high abundance of transcripts. We agree, however, that sequencing of rRNA from polysome fractions like it was done for example in *Parks et al, 2018* will test translational status of the ribosomes, and will be valuable in future work.

2. There is not a lot of discussion as to how the environment of the yeast plays a role on purifying selection. While Fig 1c discusses it briefly, the authors do not consider those VFs that exist across multiple species that may serve to differentiate how the yeast has grown in a certain environment. On page 6 lines 2-3, the authors state, “Almost half (42%) of those variants were shared across multiple isolates (Supplementary Fig. 5) but occurred at various VFs (i.e. in different proportions of rDNA copies) and thus represent variant frequency polymorphisms (VFPs; Fig. 1a).” But the authors do not state how these VFPs are distributed across the different niches. A further analysis as to the role the environment plays should be pursued. Perhaps a PCA plot colored by niche may illuminate this point.

We now added **Fig. 2** and **Supplementary Fig. 15** to discuss the role of environment in the distribution of variants and iVFPs, as well as **Supplementary Fig. 18** showing subclade distribution of iVFPs. In addition, as outlined in the response to reviewer 2, we tested an association between the most prevalent 18S and 25S variants and stress conditions used to phenotype isolates in *Peter et al., 2018* (**Supplementary Fig. 19, 20**) since the number of the variants provided enough power for the analysis. Most variants, however, are only shared among 2 to 5 isolates, which severely limits the power for identifying associations by PCA.

3. The authors make a clear distinction between VFs and VFPs, but it is not clear as to which variants are more polymorphic than others. For example, Fig S7 describes the number of shared isolates per variant, but they do not describe how polymorphic each of those sites are. How much sequence entropy/information content exists at each site?

We now calculated Shannon's entropy (H) for each rDNA position (**Supplementary File 2** and **Supplementary Fig. 7**).

4. On page 7, line 21, where the authors discuss variant pairs, several questions arise:

a. Based on Supplementary Fig. 10, it seems like the distances computed were done on a niche basis but were not done on the overall dataset. Analyses on all isolates should be reported as well.

We now added **Supplementary Fig. 12** with the distances computed on all isolates.

b. For the methods used in this paragraph, some of the conditions seemed to be arbitrarily chosen. Why does "Dxy" have to be ≤ 0.5 ? Why does "n" have to be ≥ 3 ?

Since we do not have long-read sequencing data, we chose these parameters to maximize the confidence of the computed pairs.

To do this, we empirically minimized the difference between variant iVFs (Dxy), which is a Euclidean distance between two iVFs of a variant pair. The "n" (the minimum number of isolates that contain the same pair of variants) parameter can be arbitrarily chosen, although the higher the n, the less variant pairs will be reported since the chance of sharing the same variant pair would drop. We now added this to the "Methods" section.

We believe these parameters are a good approximation since most of the variant pairs that we report have 1) high ($>80\%$) iVFs which means they must co-occur at least in a subset of rDNA copies; 2) consistent iVFs, and 3) the pipeline detected some variants that co-occurred within a short-read distance and we were able to further validate those pairs by analyzing individual sequencing reads (**Supplementary Fig. 14**).

The parameters can be chosen differently, but we did not fine-tune them since it would be beyond the scope of this study. We agree that one could take this approach further and analyze its sensitivity and specificity by varying the parameters and validating them in vivo.

c. The overarching point of this paragraph, which appears to be that variants occur in pairs, should be made more obvious.

We now expanded the paragraph.

5. In the sections titled Selection profile of the yeast ribosome and Variants in functional regions, the authors describe functional effects of the variants in the ribosome and how they may alter the geometry of the region, e.g. the GAC or the locations of such variants. The authors state that in the innermost shell, we see extremely low VFs. Yet when one peruses Supplementary File 3, one can see various variants that have high VF. A brief discussion about these positions of variance and how they may affect the decoding center or the peptidyltransferase center seems quite important and

clarifications seem warranted. It may help further elucidate which specific nucleotides have indispensable connections.

We believe this reviewer refers to shell 2 of the inner part of the ribosome, as there are no mid- and high iVFPs variants in the LSU and no high iVFPs in the SSU and only one variant with mid iVF 7.1%. However, several mid- and high frequency variants were localized to the second most inner shell 2. In the LSU, six iVFPs have intragenomic frequencies above 5%, and three of them interact with ribosomal proteins, which we now mention briefly in the manuscript. In the SSU, seven iVFPs have intragenomic frequencies above 5% in shell 2, and three of them are located in or near functionally important ribosomal regions. These are described in detail in “Variants in functional regions” (the G1112A variant is near the decoding center and increases translational fidelity in vivo experiments and other variants are within 5Å of intersubunit bridges), which suggest some of them might be either under neutral or positive selection.

6. The section rRNA analysis in the methods requires clarification:
a. Why is the expected distribution of VFPs defined to just be the fraction of nucleotides in each shell? This would appear to suggest that under a neutral selection model, 1 VFP per nucleotide is expected. I do not see how this is justified.

We justify the expected distribution by the following reasoning (added in the text as well):

- 1) The data in **Supplementary Fig. 8** suggests that the number of variants is proportional to the length of a sequence. This suggests that the longer a sequence of interest is, the higher the **chance** of a variant occurrence in this sequence;
- 2) This does not mean that one iVFP per nucleotide is expected but rather under a neutral selection model, the likelihood of an iVFP is higher with increased number nucleotides in a shell, which is given as a fraction of nucleotides in a shell over the total number of nucleotides in a subunit.

b. The chosen ranges for the VFs seem arbitrary. Why is a low VF from [0.4%-5%]? Why not [0.4%-10%]?

We empirically minimized this range based on distribution in **Fig. 1f** (see below), where the range [0.5%-5%) includes the great majority of the iVFPs of the first peak of the bimodal distribution. We followed the same logic to chose the range for the high iVFs [95%-100%] that encompasses the majority of iVFPs of the second peak in the bimodal distribution.

7. Lastly, there has been some work on how sequence differences in rDNA loci have conferred fitness benefits to organisms (*Kurylo et al Cell Reports 2018*). The authors discuss some functional changes that occur with variants, but do not address questions of fitness. A paragraph or two discussing how some of these changes might confer a fitness advantage would be engaging.

We now expanded our “*Variants in functional regions*” section as well Discussion section accordingly to address this point.

Additional questions/suggestions:

1. Next time, please set your margins in the “justify” setting in Microsoft Word for ease of viewing.

We now adjusted the margins.

2. On page 6, line 11 the authors report the mean number of isolates per variant, then proceed to compare the distributions via a non-parameterized test. For consistency, please report the medians instead.

Corrected.

3. On page 6, lines 13-16, the authors state a possible reason as to why rRNA genes have fewer isolates per variant than their ETS+ITS and NTS counterparts. Please provide a reference showing this or move this statement to the discussion.

We added a reference *Nei & Rooney, 2005*.

4. On page 6, line 22, the authors state, “...indicating regions of increased permissiveness for variation.” What does “permissiveness” mean in this context?

“Regions of increased permissiveness” means regions of lesser sequence constraint, where more sequence variation can be observed compared to the highly conserved regions/regions of stringent purifying selection. We now reworded this statement to hopefully clarify this point.

5. Please fix your usage of i.e. I.e. stands for id est which means, “that is.” E.g. stands for exempli grata which means, “for example.”

We now fixed the usage throughout the text.

6. On page 9, lines 17-19, please cite a reference or put in the discussion.

We removed this sentence from the manuscript.

7. On page 9, line 16, non-significant P-values do not need to be reported.

Removed.

8. On pages 9-10, lines 23 and 1-2, please clarify the statement, “In general, the same sequence changes presented repeatedly within clades and across related clades, while other possible changes of the same nucleotides were largely undetectable.”

We agree that the second part of the sentence is indeed confusing (we were initially referring to a variant position not a variant in one of our earlier manuscript drafts and forgot to change it later) and we edited the sentence by removing the second part and changing “sequence changes” to “variants”.

9. On page 11, line 16, why did you not do a p-value correction here as you did previously?

Thank you for pointing this out. We now performed a p-value correction for multiple hypothesis testing there and everywhere in the text/figures where relevant (more than two groups being compared).

Reviewers' Comments:

Reviewer #1:

Remarks to the Author:

The rewriting of the Results, modification/addition of figures, expansion of the Introduction and Discussion, and the additional experiments realised by the authors have certainly improved the clarity of the study. In the revised version, the authors have addressed and thus clarified most of my (and other reviewers') concerns/comments and I thank them for the amount of work they have done and their attention to details.

Questions/Comments

1-While a new description of VFs and VFps (iVFs and iVFps in the revised version) has been given, iVFP values do not just represent intragenomic variant frequency polymorphisms but the sum of all intragenomic variant frequency polymorphisms observed at one nucleotide position amongst the 918 isolates (e.g., Fig 1c, iVFP, n, scale 0 to 925). The notion of sum is mentioned page 34, lane 7 (Legend, figure 1) and figure 1a also somewhat displays it but that deserves to be clarified/mentioned in the text as well. In that regard, lanes 8-11 of page 6 should be reformulated.

That clarification and precision of the terminology are essential especially as the 'high' iVFPs values are used by the authors to "...show that rDNA copies in this yeast are far from homogeneous,..." (page 2- lanes5-6); "...reveal pervasive heterogeneity in rDNA sequences within and between genomes of the same species..." (page 9, lane1-2); in order to overall argue "...against rapid sequence homogenization of rDNA copies by concerted evolution." (page 12, lane7-8, as example). Despite of the high quality of the work delivered, I am still not convinced that the results strongly argue against the concerted evolution phenomenon that for the best of my knowledge, does not consider the evolution of repeated genes in one species but does consider the evolution of repeated genes in one species compared to their evolution in closely related species.

Minor points

2-page 3, lanes 10-11, see phrasing. "By contrast, but how much rDNA..."

3-page 12 lane 3, "For variants in rRNA genes, the wide fluctuation..."

4-page 14, lanes 19-20. Check numbers given in the text versus numbers in Figure 5d. See also page 39, lane 11 (Legend figure 5): "Right; three variants have iVFPs with iVFs both below and above 6 %...". Figure 5d (Venn diagram) shows two variants not three.

5-page 19 lanes 16-17, see phrasing. "Regulatory mechanisms that activate these dormant rRNA genes or generally increase rRNA transcription, thus provide a substantial..."

Reviewer #2:

Remarks to the Author:

I'm fully happy with this revised version and support publication in its current form.

Reviewer #3:

Remarks to the Author:

The revised manuscript by Sultanov and Hochwagen appears to have been substantially improved by this review process. Their revised manuscript contains new data supporting their conclusions (in particular the rRNA variant analyses), new analyses and new figures that further support their contention that their observations stand in stark contrast to a purely concerted evolution model.

While there are a few statements (listed below) that may require additional clarification, in my view this work deserves to move forward to publication in Nature Communications.

1] what does human-associated mean exactly?

2] how do the authors know that disease-associated rDNA/rRNA variants don't exist? Has this been looked into with the scrutiny needed to make this claim? I would recommend stating that it should be looked into based on the presented data and data from other groups, including the referenced works of Parks, Kurylo et al.

We thank the editor and the reviewers for evaluating our revised manuscript. We modified the text to improve clarity and address questions and comments raised by the reviewers.

REVIEWER COMMENTS

Reviewer #1 (Remarks to the Author):

The rewriting of the Results, modification/addition of figures, expansion of the Introduction and Discussion, and the additional experiments realized by the authors have certainly improved the clarity of the study. In the revised version, the authors have addressed and thus clarified most of my (and other reviewers') concerns/comments and I thank them for the amount of work they have done and their attention to details.

We thank the reviewer for their positive evaluation of this version and their constructive critique, and we are glad our revision was helpful.

Questions/Comments

1-While a new description of VFs and VFps (iVFs and iVFPs in the revised version) has been given, iVFP values do not just represent intragenomic variant frequency polymorphisms but the sum of all intragenomic variant frequency polymorphisms observed at one nucleotide position amongst the 918 isolates (e.g., Fig 1c, iVFP, n, scale 0 to 925). The notion of sum is mentioned page 34, lane 7 (Legend, figure 1) and figure 1a also somewhat displays it but that deserves to be clarified/mentioned in the text as well. In that regard, lanes 8-11 of page 6 should be reformulated.

That clarification and precision of the terminology are essential especially as the 'high' iVFPs values are used by the authors to "...show that rDNA copies in this yeast are far from homogeneous,..." (page 2- lanes5-6); "...reveal pervasive heterogeneity in rDNA sequences within and between genomes of the same species..." (page 9, lane1-2); in order to overall argue "...against rapid sequence homogenization of rDNA copies by concerted evolution." (page 12, lane7-8, as example).

The iVFP by itself is a variant at a given frequency. If the variant is shared among two isolates, the total **number** of iVFPs for this variant is two. We indeed analyze the sum of all iVFPs, e.g. when we compare differences between NTS, ETS+ITS, and rRNA elements (**Fig. 1 e-g**), as a proxy for total variation (more variable regions will have more iVFPs), but we explicitly label axes as "iVFP, n" (i.e. "the number of iVFPs").

Regarding "at one nucleotide position": a variant is defined by **both** sequence and position. For example, A100G and A100C are two different variants, and A100G can have 3 iVFPs whereas A100C can have 5 iVFPs. But since they share the same position, they will both contribute to the height of a histogram bar at position 100 in **Fig. 1c** (we now clarified this aspect in the figure legend).

Despite of the high quality of the work delivered, I am still not convinced that the results strongly argue against the concerted evolution phenomenon that for the best of my knowledge, does not consider the evolution of repeated genes in one species but does consider the evolution of repeated genes in one species compared to their evolution in closely related species.

The reviewer raises an important point that we now clarified further in the introduction and discussion. Specifically, the data in our manuscript, argue against several key assumptions used

to describe the **results** of concerted evolution (not against the phenomenon itself) in the rDNA of a species. These assumptions are homogeneous rDNA copies, no bias in the variant distribution, and rapid fixation/elimination of variants. Our analyses also investigate the previously overlooked role and strength of selection in shaping rDNA sequence diversity, which acts along the concerted mode of rDNA evolution. Although concerted evolution indeed has been defined as the phenomenon where paralogous genes are more closely related to one another than to the corresponding orthologs in closely related species, the same terminology has also been used to define evolution at the level of individual genomes (e.g. Ganley & Kobayashi, 2007, 2011 [PMIDs: 17200233, 21546356]) and populations (e.g. Zimmer et al., 1980 [PMID: 6929543]), when paralogs share synapomorphic changes (e.g. new mutations spread within an array of duplicated genes through gene conversion). For example, Zimmer et al., 1980 [PMID: 6929543] wrote: “*We stress the importance of recognizing that this process takes place in populations*”). Additionally, Elder & Turner, 1995 (PMID: 7568673) argued: “*There is no a priori reason to consider either molecular drive or concerted evolution as species-level phenomena. In order for them to have effects at the species level, they must begin to operate in naturally occurring populations ... If any conclusions about the evolutionary significance of molecular drive are to be reached, data that report the divergence of repetitive DNAs at the population level are needed. It is only by examining how repetitive DNAs vary within and among populations that any real understanding of molecular drive and concerted evolution will be achieved.*” We now added this reference in the manuscript.

Minor points

2-page 3, lanes 10-11, see phrasing. “By contrast, but how much rDNA...”

Corrected

3-page 12 lane 3, “For variants in rRNA genes, the wide fluctuation...”

We’ve changed “fluctuation” to “range”

4-page 14, lanes 19-20. Check numbers given in the text versus numbers in Figure 5d. See also page 39, lane 11 (Legend figure 5): “Right; three variants have iVFPs with iVFs both below and above 6 %...”. Figure 5d (Venn diagram) shows two variants not three.

Thank you for spotting this, we have corrected the numbers.

5-page 19 lanes 16-17, see phrasing. “Regulatory mechanisms that activate these dormant rRNA genes or generally increase rRNA transcription, thus provide a substantial...”.

We re-phrased the sentence and broke it into two statements.

Reviewer #2 (Remarks to the Author):

I'm fully happy with this revised version and support publication in its current form.

We thank the reviewer for their expertise and assessment of our work.

Reviewer #3 (Remarks to the Author):

The revised manuscript by Sultanov and Hochwagen appears to have been substantially improved by this review process. Their revised manuscript contains new data supporting their conclusions (in particular the rRNA variant analyses), new analyses and new figures that further support their contention that their observations stand in stark contrast to a purely concerted evolution model.

While there are a few statements (listed below) that may require additional clarification, in my view this work deserves to move forward to publication in Nature Communications.

We thank the reviewer for their positive evaluation of our work and their additional suggestions.

1] what does human-associated mean exactly?

These are “human body-associated” isolates. We now clarified this in the manuscript and the x-axis labels in **Fig. 2** (“Human body” instead of “Human”).

2] how do the authors know that disease-associated rDNA/rRNA variants don't exist? Has this been looked into with the scrutiny needed to make this claim? I would recommend stating that it should be looked into based on the presented data and data from other groups, including the referenced works of Parks, Kurylo et al.

Yes, this is exactly what we meant with our statement. It is quite likely that there will be rDNA/rRNA-disease variants, but those have been largely overlooked because rDNA is not analyzed in genotype-phenotype association studies, especially for low-frequency variants. We reworded this statement to indicate that, looking into such associations will be a very important endeavor.